# Elimination of fukutin reveals cellular and molecular pathomechanisms in muscular dystrophy-associated heart failure

Yoshihiro Ujihara[1,2,4], Motoi Kanagawa[3], Satoshi Mohri[1,2], Satomi Takatsu[1], Kazuhiro Kobayashi[3], Tatsushi Toda[3], Keiji Naruse [1] & Yuki Katanosaka [1]*

Heart failure is the major cause of death for muscular dystrophy patients, however, the molecular pathomechanism remains unknown. Here, we show the detailed molecular pathogenesis of muscular dystrophy-associated cardiomyopathy in mice lacking the fukutin gene (*Fktn*), the causative gene for Fukuyama muscular dystrophy. Although cardiac *Fktn* elimination markedly reduced α-dystroglycan glycosylation and dystrophin-glycoprotein complex proteins in sarcolemma at all developmental stages, cardiac dysfunction was observed only in later adulthood, suggesting that membrane fragility is not the sole etiology of cardiac dysfunction. During young adulthood, *Fktn*-deficient mice were vulnerable to pathological hypertrophic stress with downregulation of Akt and the MEF2-histone deacetylase axis. Acute *Fktn* elimination caused severe cardiac dysfunction and accelerated mortality with myocyte contractile dysfunction and disordered Golgi-microtubule networks, which were ameliorated with colchicine treatment. These data reveal fukutin is crucial for maintaining myocyte physiology to prevent heart failure, and thus, the results may lead to strategies for therapeutic intervention.

[1] Department of Cardiovascular Physiology, Graduate School of Medicine, Dentistry and Pharmaceutical Sciences, Okayama University, Shikata-cho 2-7-1, Okayama City, Okayama 7008558, Japan. [2] Department of Physiology, Kawasaki Medical School, Matsushima 577, Kurashiki City, Okayama 7010192, Japan. [3] Division of Neurology/Molecular Brain Science, Kobe University Graduate School of Medicine, Kobe 650-0017, Japan. [4] Present address: Department of Electrical and Mechanical Engineering, Graduate School of Engineering, Nagoya Institute of Technology, Nagoya 466-8555, Japan. *email: ytanigu@md.okayama-u.ac.jp

The dystrophin-glycoprotein complex (DGC) links the intracellular cytoskeleton to the extracellular basement membrane, thereby providing structural support for the sarcolemma[1,2]. Within this complex, dystroglycan (DG) links intracellular dystrophin to the extracellular matrix[3]. As heavily glycosylated α-DG binds tightly and with high affinity to laminin[4], a glycosylation deficiency can dissolve the DGC–extracellular matrix connection. Mutations of dystrophin and the transmembrane sarcoglycans result in Duchenne/Becker and limb girdle muscular dystrophies, respectively[4–7]. Many patients with muscular dystrophies, particularly those with defects in the DGC, develop cardiomyopathies with chamber dilation and myocardial dysfunction[8–12], which is the major cause of death in these patients.

Recently, the full functional glycan structure of α-DG required for its ligand binding activities was identified[13–15], providing insight into the pathogenesis and therapeutic strategies for α-DG-associated diseases, including Fukuyama-type congenital muscular dystrophy (FCMD)[16], muscle–eye–brain disease[17], Walker–Warburg syndrome[18], congenital muscular dystrophy types 1C and 1D[19], and limb girdle muscular dystrophy type 2I[20]. Although the causative genes for these diseases differ, one biochemical feature that is common among them is abnormal glycosylation of α-DG. Thus, these conditions are considered α-DGpathies. FCMD is caused by mutation of the fukutin gene (FKTN)[16], which encodes a Golgi-based ribitol phosphate transferase that catalyzes the biosynthesis of tandem ribitol phosphate structures on α-DG by sequential action with fukutin-related protein (FKRP)[13]. Mutations in FKTN and FKRP-associated α-DGpathy are both associated with cardiomyopathies[21,22]. However, the detailed molecular mechanisms for cardiac pathogenesis in these conditions remain unknown.

In this study, we show a crucial role for FKTN in the maintenance of myocyte structure and function using cardiac-specific Fktn knockout mice. The results from this study improve our understanding of the pathomolecular mechanism underlying muscular dystrophy-associated heart failure.

## Results

**Cardiac changes in MCK-*Fktn*-cKO (*Fktn^flox/flox*; *MCK-Cre^+/−*) mice.** To examine the effect of FKTN deficiency, we crossed *Fktn^flox/flox* (floxed) mice with a transgenic line expressing Cre recombinase under the control of the MCK promoter[23]. The reduced expression of FKTN and DGC proteins in the hearts of these MCK-*Fktn*-cKO mice was confirmed by immunoprecipitation and immunoblotting (Supplementary Fig. 1a, b). The glycosylated form of α-DG and DGC immunoreactivity increased in the sarcolemma at 24 weeks after birth in normal control mice (Fig. 1a), suggesting that the physiological contribution of DGC proteins in the heart increases 6 months after birth. Notably, FKTN deficiency not only resulted in a reduction of α-DG glycosylation but also impaired the expression of DGC proteins in the sarcolemma at all developmental stages (Fig. 1a). We next examined global cardiac morphology and function in MCK-*Fktn*-cKO mice at different time points after birth. Cardiac gross morphology (Fig. 1b) and the heart-to-body weight ratio (Fig. 1c) were maintained in mice with *Fktn* deficiency. However, the cross-sectional areas of cardiomyocytes and fibrosis were increased in MCK-*Fktn*-cKO hearts 24–48 weeks after birth (Fig. 1d–f). These mice also showed cardiac dysfunction and chamber dilation during diastole at this age (Fig. 1g, h). Only one or a few damaged areas were detected by anti-mouse IgG in 48-week-old MCK-*Fktn*-cKO mice (Supplementary Fig. 2). The 48-week-old heterozygous (*Fktn^flox/+*; *MCK-Cre^+/−*) mice showed no abnormalities in overall cardiac morphology and function

(Supplementary Fig. 3). Thus, *Fktn* deficiency leads to pathological cardiac remodeling in 24–48-week-old mice.

**Impaired myocyte function in 10-month-old MCK-*Fktn*-cKO mice.** The subcellular structure of cardiomyocytes and the localization of Ca$^{2+}$ regulatory proteins are well suited to their cellular functions[24]. Well-ordered patterns of immunofluorescence were observed for L-type Ca$^{2+}$ channels (LTCC), Na$^+$/Ca$^{2+}$ exchanger 1 (NCX1), and ryanodine receptor 2 (RyR2) in normal hearts (Fig. 2a, left) but not in 10-month-old MCK-*Fktn*-cKO hearts (Fig. 2a, right), suggesting a defect in intracellular Ca$^{2+}$ handling. The myofilaments needed to generate force were extensively disorganized in the MCK-*Fktn*-cKO hearts (Fig. 2b), and cardiomyocytes from these hearts showed reduced shortening (Fig. 2c) and impaired Ca$^{2+}$ handling during excitation–contraction (E–C) coupling (Fig. 2d). This abnormality in Ca$^{2+}$ cycling was most evident at 5 mM, the concentration at which *Fktn*-deficient myocytes also showed reduced amplitude (Fig. 2e), longer time constants (i.e., slower decay speed) (Fig. 2f), and increased time to peak (Fig. 2g). As a result, Ca$^{2+}$ contents were lower in the sarcoplasmic reticula (SRs) of MCK-*Fktn*-cKO myocytes than in those of floxed controls (Fig. 2h). Thus, cardiac dysfunction in adult MCK-*Fktn*-cKO mice occurred at the cardiomyocyte level.

**Hypertrophic responses of hearts from MCK-*Fktn*-cKO mice.** As mentioned above, young-adult MCK-*Fktn*-cKO mice did not exhibit significant cardiac dysfunction under physiological conditions (Fig. 1). Consistently, isolated cardiomyocytes from 10-week-old MCK-*Fktn*-cKO mice showed normal contractility and Ca$^{2+}$ handling during E–C coupling (Supplementary Fig. 4). To investigate the effect of FKTN deficiency under stress conditions, we examined cardiac structure, function, and intracellular signaling after inducing various hemodynamic stresses. Control mice showed incremental changes in heart weight-to-body weight ratios (Fig. 3a, b) and cross-sectional areas of cardiomyocytes (Fig. 3c) without increased fibrosis (Fig. 3d) after administration of the adrenergic receptor agonists isoproterenol and phenylephrine as well as after prolonged pressure overload via 2-week thoracic aortic constriction. Conversely, these treatments caused chamber dilation and fibrosis with severe cardiac dysfunction in the hearts of MCK-*Fktn*-cKO mice (Fig. 3a, d, e). These observations suggest that *Fktn*-deficient hearts are vulnerable to hypertrophic stress and that FKTN in the heart is essential for compensatory hypertrophic responses against hemodynamic stress.

Next, in order to understand the molecular basis underlying vulnerability, we analyzed the changes in intracellular signaling in the hearts of *Fktn*-deficient mice treated with isoproterenol for 2 weeks. It is well known that Akt and Ca$^{2+}$ calmodulin-dependent protein kinase II (CamKII) phosphorylation mediate myocyte survival and apoptotic/necrosis pathways[25–27], respectively. We found that isoproterenol increased the phosphorylation of Akt in control hearts but not in *Fktn*-deficient hearts, which instead showed increased phosphorylation of CamKII, suggesting a downregulation of survival and upregulation of apoptosis/necrosis signaling. Thus, *Fktn*-deficient hearts showed an impaired hypertrophic response to stress.

It is also known that activation of protein kinase D (PKD) leads to the phosphorylation of histone deacetylases (HDACs) in myocyte pathological remodeling[28–30]. Whereas isoproterenol treatment slightly induced the phosphorylation of PKD in control hearts, basal levels of PKD phosphorylation in the absence of isoproterenol were already significantly higher in MCK-*Fktn*-cKO hearts than in controls (Fig. 3f). Similarly, the phosphorylation of

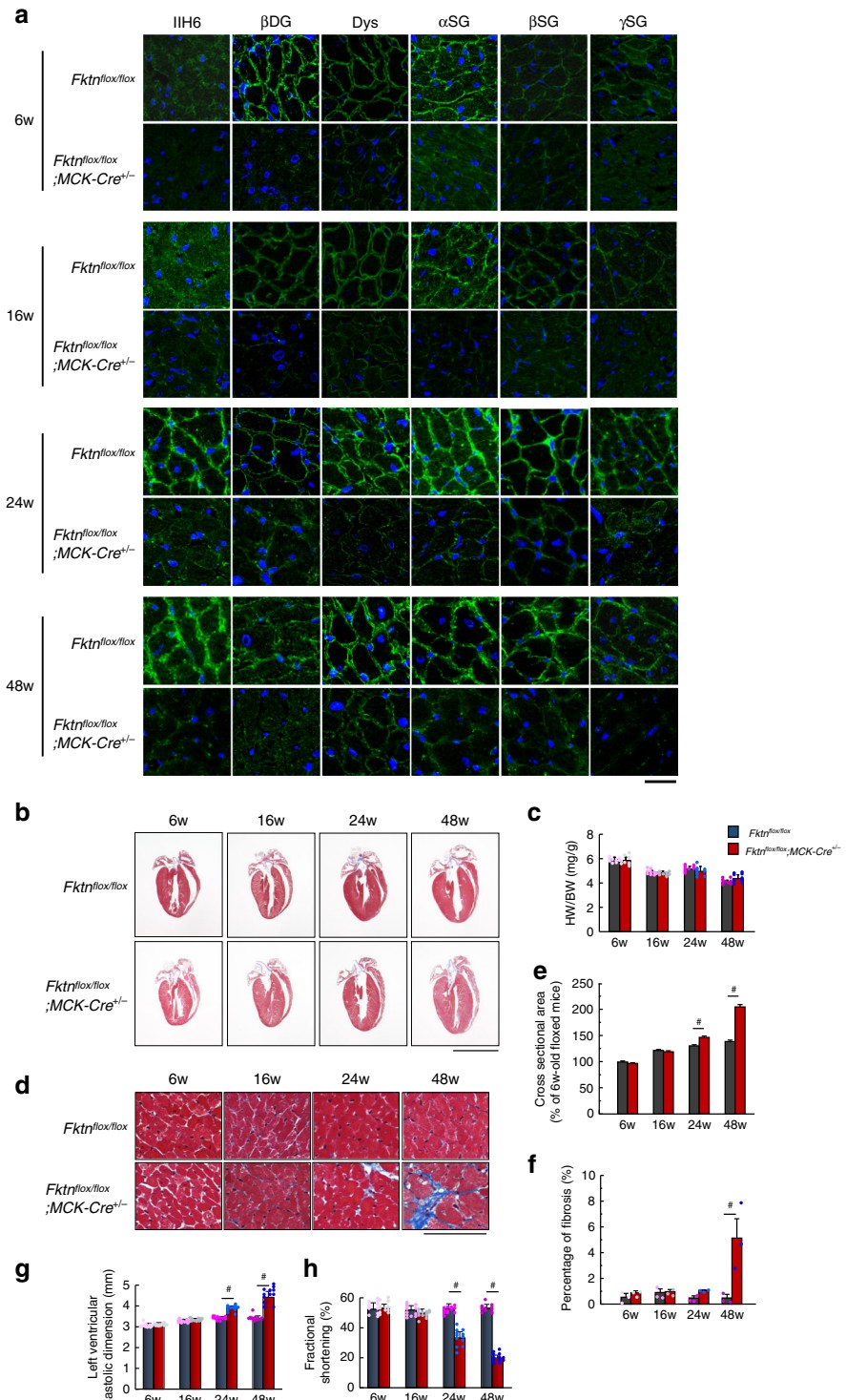

**Fig. 1 Cardiac change in MCK-*Fktn*-cKO (*Fktn*^flox/flox^; *MCK-Cre*+/−) mice. a** Representative immunofluorescent images of DGC complex and glycosylation form of α-DG in control (floxed) and MCK-*Fktn*-cKO hearts. Anti-IIH6 (glycosylated form of αDG), βDG, dystrophin (Dys), αSG, βSG, and γSG antibodies (green) and DAPI (blue). Scale bar, 10 μm. **b** Cardiac morphology in floxed and MCK-*Fktn*-cKO mice. Scale bar, 5 mm. **c** Heart weight-to-body weight ratios (HW/BW) ($n = 9$ mice per group). **d** Masson's trichrome staining of the left ventricle. Scale bar, 50 μm. Cross-sectional areas (6w, $n = 259$ and 268 cells; 16w, $n = 303$ and 311 cells; 24w, $n = 266$ and 287 cells; 48w, $n = 283$ and 256 cells measured from 3 hearts per group.) (**e**) and fibrosis percentages ($n = 3$ mice per group) (**f**) from paraffin sections of left ventricles. Echocardiographic parameters of left ventricle diastolic dimension (**g**) and fractional shortening (**h**) in floxed and MCK-*Fktn*-cKO mice ($n = 12$ mice per group). Data are means ± s.e.m; #$P < 0.05$ between indicated groups based on Student's t-tests.

HDAC9 in control hearts was increased by isoproterenol treatment, while the basal phosphorylation in MCK-*Fktn*-cKO hearts was already extremely high. These data indicate that *Fktn* elimination enhanced the PKD signaling pathways, provoking HDAC9 nucleocytoplasmic shuttling under physiological conditions. Therefore, *Fktn* elimination accelerates the progression from compensated cardiac hypertrophy to heart failure under hemodynamic stress conditions.

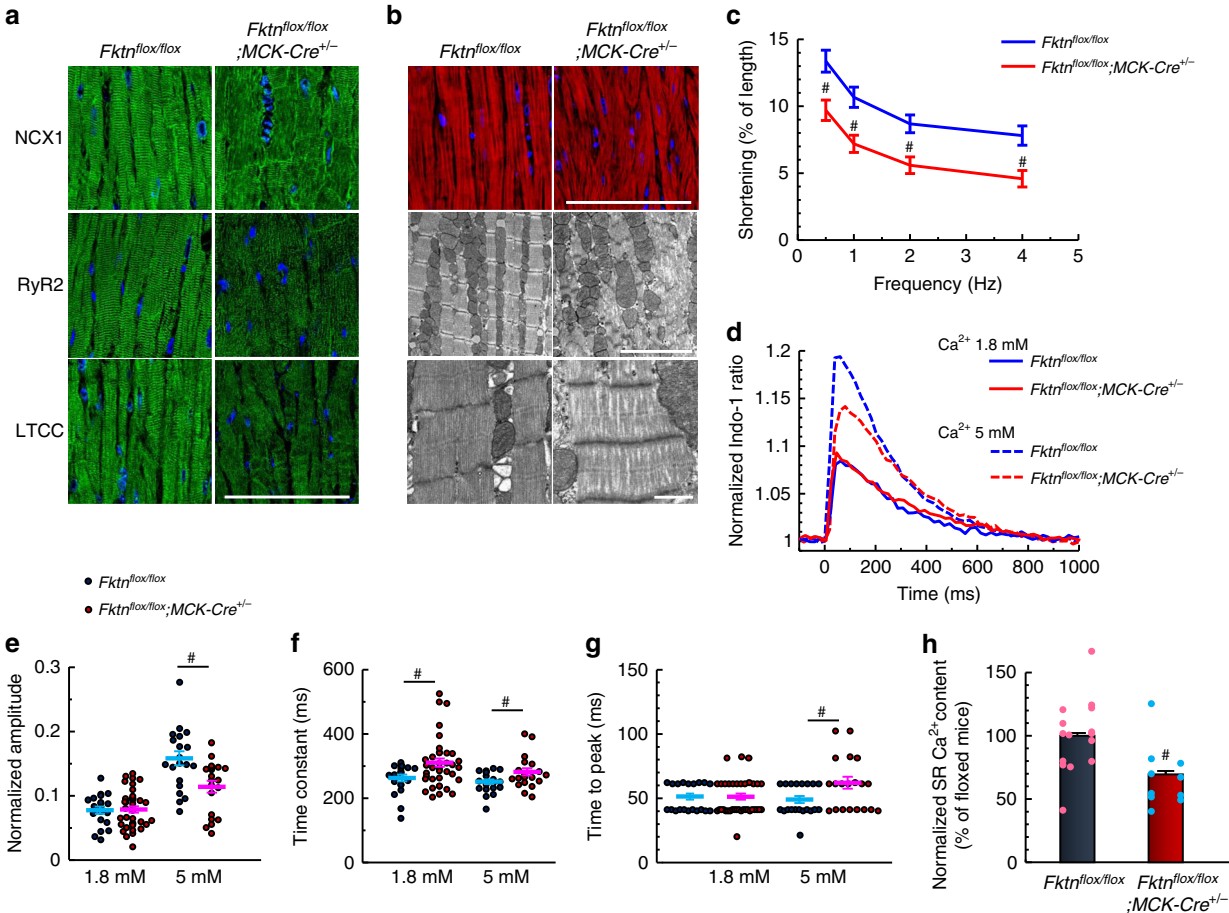

**Fig. 2 Impaired myocyte function in 10-month-old MCK-*Fktn*-cKO (*Fktn*^flox/flox^; *MCK-Cre*^+/−^) mice. a** Representative immunostaining for NCX1, RyR2, and LTCC (DAPI counterstain, blue) in hearts of 10-month-old mice. Scale bar, 100 μm. **b** Representative immunofluorescence (top: phalloidin, red; DAPI, blue (scale bar, 100 μm)) and electron microscopy (scale bar, 5 μm (middle) and 1 μm (lower)) images of myofilaments. **c** Frequency-dependent shortening of cardiomyocytes ($n = 16$ and 20 cells measured from 4 and 5 hearts, respectively). #$P < 0.05$ vs. floxed mice based on Student's *t*-tests. (**d**) Indo-1 fluorescence in single cardiomyocytes stimulated at 1 Hz (1.8 mM $Ca^{2+}$, $n = 20$ and 37 cells from 3 and 4 hearts, respectively; 5 mM $Ca^{2+}$, $n = 20$ cells from 3 hearts per group). Normalized peak amplitudes (**e**), decay time constants (obtained by fitting to the decline phase) (**f**), and times to peak (**g**) of $Ca^{2+}$ transients. **h** Estimation of sarcoplasmic reticulum (SR) $Ca^{2+}$ content ($n = 15$ and 10 cells from 3 and 2 hearts, respectively). #$P < 0.05$ between indicated groups based on Student's *t*-tests.

**Impaired hypertrophic response in MCK-*Fktn*-cKO cardiomyocytes.** To more directly examine the hypertrophic responses of MCK-*Fktn*-cKO cardiomyocytes, we analyzed cell growth during culture with or without phenylephrine for 48 h. We confirmed successful Cre-lox recombination and expression of Cre recombinase in the nucleus in cultured neonatal cardiomyocytes from MCK-*Fktn*-cKO mice (Supplementary Fig. 5). We also confirmed reduction of FKTN protein in 1-d-old MCK-*Fktn*-cKO mice (Supplementary Fig. 5). In the presence of phenylephrine, cardiomyocytes from control mice showed enhanced sarcomere organization and upregulation of NCX1 expression (Fig. 4a), which are signs of myofibril maturation and intracellular $Ca^{2+}$ handling during E–C coupling[24]. By contrast, MCK-*Fktn*-cKO cardiomyocytes treated with phenylephrine had immature and tangled myofilaments (Fig. 4b) and showed no increase in cell area (Fig. 4c). We confirmed that all cKO myocytes treated with phenylephrine were alive using a LIVE/DEAD assay (Fig. 4d). Remarkably, cKO myocytes did not show spontaneous hypertrophy (at 48 h in the absence of phenylephrine; Fig. 4c). These myocytes exhibited no sarcomere formation, cell–cell interaction (Fig. 4a, upper right panels), or spontaneous beating, suggesting that *Fktn* deficiency also affects myocyte maturation in cKO cells. $Ca^{2+}$ content in the SRs of control cardiomyocytes increased

during culture with phenylephrine (Fig. 4e), whereas no such increase was observed in cardiomyocytes from MCK-*Fktn*-cKO mice. However, caffeine-induced release was still present, suggesting that these cells had functions characteristic of cardiomyocytes. As a control, mixed fibroblasts in the cultured myocyte preparation showed no response to caffeine (Fig. 4e). These observations suggest that *Fktn* elimination in myocytes impairs hypertrophic responses.

Using these cell models, we examined the subcellular localization of MEF2, which acts with class IIa HDACs, such as HDAC9, in myocyte differentiation and compensatory hypertrophic responses[31]. MEF2 underwent nuclear translocation in control cardiomyocytes but not in MCK-*Fktn*-cKO cardiomyocytes cultured with 10% fetal calf serum (FCS) or 10 μM phenylephrine for 48 h (Fig. 5a). In control cardiomyocytes, serum (as well as adrenergic agonists) induced the nuclear export of HDAC9, which relieves transcriptional repression[28]. This phenomenon was not observed in MCK-*Fktn*-cKO cardiomyocytes (Fig. 5b). These observations suggest that MEF2 and HDAC9 translocation are impaired in FKTN-deficient cardiomyocytes.

Phosphatidylinositol 4-phosphate (PI4P) in the Golgi apparatus is required for PKD activation in pathological hypertrophic

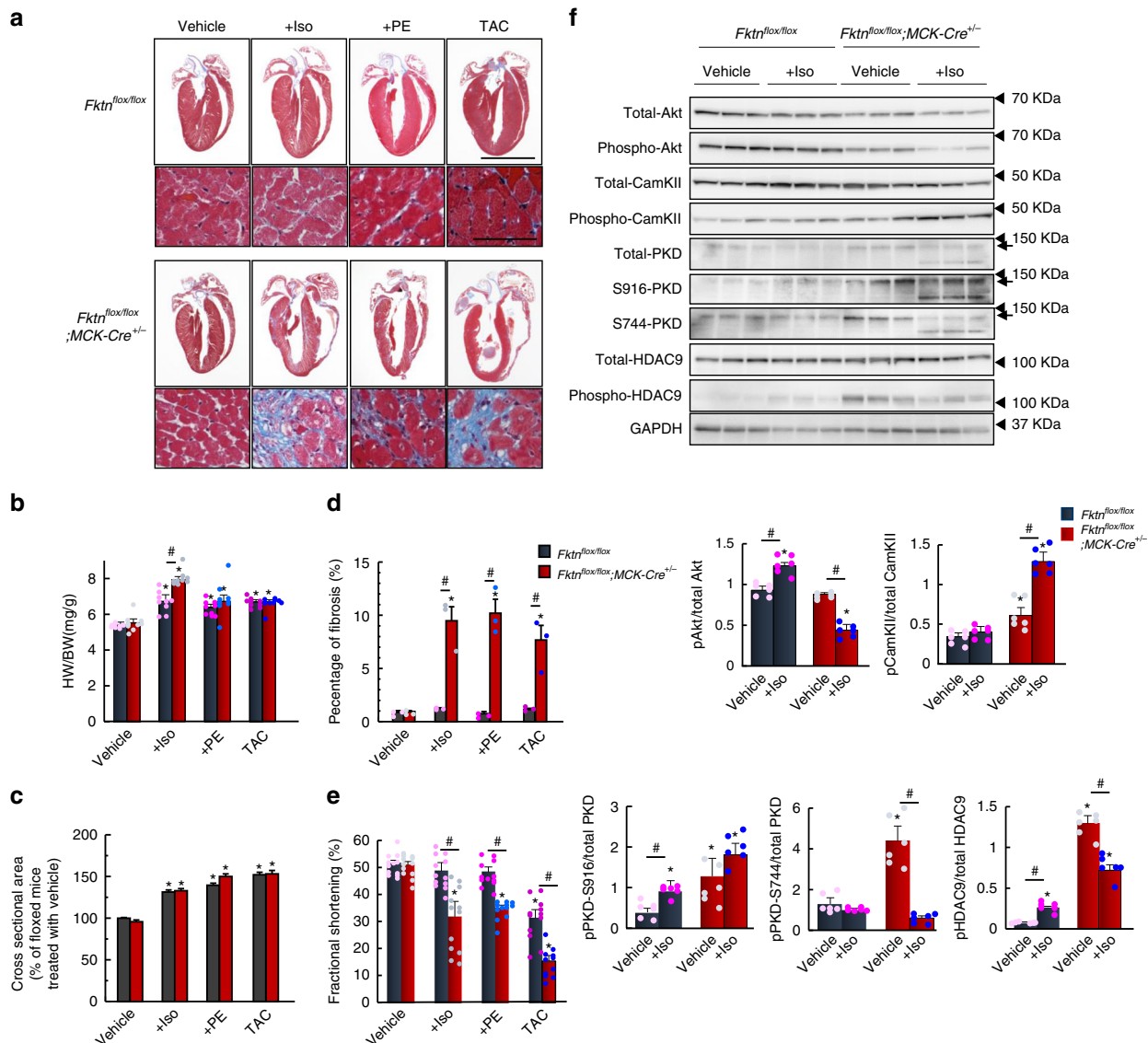

**Fig. 3 Hypertrophic responses of 10-week-old MCK-*Fktn*-cKO (*Fktn*<sup>flox/flox</sup>; *MCK-Cre*<sup>+/−</sup>) hearts. a** Cardiac morphology and histology following isoproterenol (Iso) or phenylephrine (PE) administration or thoracic aortic constriction (TAC) for 2 weeks in control (floxed) and MCK-*Fktn*-cKO mice. Scale bar, 5 mm (top) and 50 μm (bottom). **b** Heart weight-to-body weight ratios (HW/BW) ($n = 9$ mice per group). Cross-sectional areas (vehicle, $n = 252$ cells; +Iso, $n = 283$ cells; +PE, $n = 249$ cells; TAC, $n = 264$ cells from 3 control (floxed) mice. Vehicle, $n = 315$ cells; +Iso, $n = 250$ cells; +PE, $n = 229$ cells; TAC, $n = 274$ cells from 3 MCK-*Fktn*-cKO mice) (**c**) and fibrosis percentages ($n = 3$ mice per group) (**d**) from paraffin sections of left ventricles. **e** Fractional shortening in hearts of treated and untreated control ($n = 12$ mice per group). *$P < 0.05$ vs. vehicle-treated floxed mice; #$P < 0.05$ between indicated groups based on Tukey–Kramer tests. **f** Representative immunoblots for total and phosphorylated Akt, CamKII, PKD, and HDAC9 levels in control and MCK-*Fktn*-cKO mice after Iso (or vehicle) treatment. GAPDH was used for a loading control ($n = 6$ per group). *$P < 0.05$ vs. vehicle-treated floxed mice; #$P < 0.05$ between indicated groups based on Tukey–Kramer tests.

remodeling[32,33]. *Fktn*-deficient myocytes showed abnormal accumulation of PI4P (Fig. 5c) and surprisingly, fragmentation of the Golgi apparatus under basal conditions, as revealed by staining with the cis-Golgi marker protein GM130 (Fig. 5d). These phenotypes may be indicative of PKD hyperactivation, which subsequently impairs the MEF2-HDAC axis following hypertrophic stimulation. Furthermore, PKD hyperactivation also may be associated with Golgi fragmentation. Treatment of HEK cells with the marine sponge metabolite ilimaquinone (a PKD activator) induced the vesiculation of the Golgi, as observed via GM130 staining (Fig. 5e, top middle). Alternatively, *Fktn* elimination per se may affect Golgi structure. Remarkably, we found that untreated *FKTN*-deficient HEK cells showed an elongated pattern of GM130 staining, suggesting that FKTN is

involved in maintaining the structure of the Golgi apparatus (Fig. 5e, bottom left). In support of this notion, treatment with ilimaquinone or the PKD inhibitor CID755673 did not result in further disruption (Fig. 5e, bottom panels). In HAP cells, the structure of the Golgi apparatus was changed in an FKTN-dependent manner (Fig. 5f). These observations suggest the possibility that FKTN contributes to the maintenance of the Golgi architecture in these cells.

**Acute elimination of *Fktn* results in severe cardiac dysfunction.** Although FKTN protein is likely involved in the maintenance of myocytes, we did not observe structural and functional defects in cardiomyocytes from young-adult mice. To examine the direct

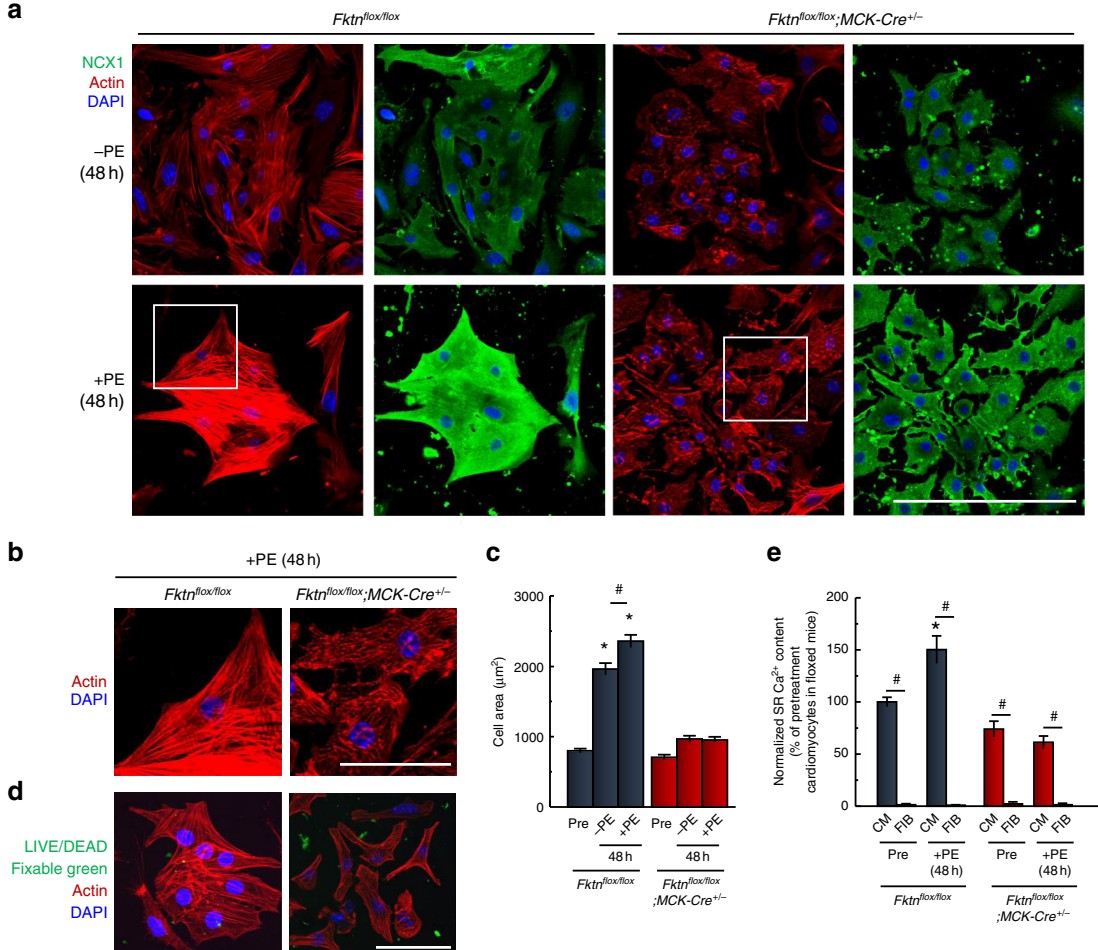

**Fig. 4 Impaired hypertrophic responses of MCK-*Fktn*-cKO (*Fktn^flox/flox*; *MCK-Cre^+/−*) cardiomyocytes. a** Representative immunofluorescence images of neonatal cardiomyocytes cultured with or without phenylephrine (PE) for 48 h (NCX1, green; phalloidin, red; DAPI, blue). Scale bar, 200 μm. **b** Magnified view of white squares in **a**. Scale bar, 50 μm. **c** Change in cell area of PE-treated cardiomyocytes (Pre, $n = 106$ cells; −PE, $n = 98$ cells; +PE, $n = 96$ cells from 10 hearts isolated from floxed mice. Pre, $n = 59$ cells; −PE, $n = 87$ cells; +PE, $n = 113$ cells from 10 hearts isolated from MCK-*Fktn*-cKO mice.). *$P <$ 0.05 vs. cardiomyocytes before treatment with PE (Pre) isolated from floxed or MCK-*Fktn*-cKO mice and MCK-*Fktn*-cKO myocytes treated with or without PE; #$P < 0.05$ between indicated groups based on Tukey–Kramer tests. **d** Viability of cardiomyocytes from floxed or MCK-*Fktn*-cKO hearts. Confocal microscope images, LIVE/DEAD assays for myocytes cultured for 48 h. Scale bar, 100 μm. **e** Estimation of SR Ca$^{2+}$ content (Pre-CM, $n = 52$ cells; Pre-FB, $n = 13$ cells; +PE-CM, $n = 50$ cells; +PE-FB, $n = 26$ cells from 10 hearts isolated from floxed mice. Pre-CM, $n = 33$ cells; Pre-FB, $n = 46$ cells; +PE-CM, $n = 45$ cells; +PE-FB, $n = 23$ cells from 10 hearts isolated from MCK-*Fktn*-cKO mice.). CM cardiomyocytes, FIB fibroblasts. *$P < 0.05$ vs. cardiomyocytes before treatment with PE (Pre) isolated from floxed mice; #$P < 0.05$ between indicated groups based on Tukey–Kramer tests.

effect of *Fktn* elimination in young-adult mice, we generated temporally controlled cardiomyocyte-specific *Fktn*-deficient [αMHC-MerCreMer (MCM)-*Fktn*-cKO] mice (Supplementary Fig. 6). The treatment of 10-week-old αMHC-MCM-*Fktn*-cKO (*Fktn^flox/flox*; *αMHC-MCM^+/−*) mice for 4 d with tamoxifen-induced chamber dilation and severe cardiac dysfunction (Fig. 6a, b). Surprisingly, ~50% of these mice died within 1 week after initiating tamoxifen treatments (Fig. 6c), suggesting an indispensable role for FKTN in the working heart. *Fktn^flox/+*; *αMHC-MCM^+/−* (hetero) mice showed no abnormalities in overall cardiac structure and function or mortality rate after tamoxifen treatment (Supplementary Fig. 7). As previously mentioned (Fig. 1a), the glycosylation of α-DG (as detected with the IIH6 antibody) increases with age, such that levels are barely detectable in hearts from 10-week-old mice. The hearts of tamoxifen-treated αMHC-MCM-*Fktn*-cKO mice showed normal DGC protein localization and levels of glycosylated α-DG (Supplementary Fig. 8); this suggests that cardiac dysfunction at this stage (4 d after tamoxifen injection) is independent of α-DG glycosylation

and DGC proteins in the sarcolemma. Cardiomyocytes isolated from tamoxifen-treated αMHC-MCM-*Fktn*-cKO mice had severely impaired contractility and Ca$^{2+}$ handling during E–C coupling with reduced amplitudes and delayed times to peak of electrically evoked Ca$^{2+}$ transients compared with those from vehicle-treated controls (Fig. 6d–h).

**Subcellular changes in αMHC-MCM-*Fktn*-cKO cardiomyocytes.** We next investigated how *Fktn* elimination affects contractility and Ca$^{2+}$ handling during E–C coupling, particularly via the structure of T-tubules and myofilaments and expression of Ca$^{2+}$ regulatory proteins. We analyzed the T-tubule structure and found it was severely disordered in *Fktn*-deficient cardiomyocytes (Fig. 7a). The myocardia of tamoxifen-treated αMHC-MCM-*Fktn*-cKO mice also showed subdivided subcellular myofilaments and disorganized Z-line structures (Fig. 7b, black arrowheads). Thus, *Fktn* elimination directly leads to structural and functional cardiomyocyte defects, especially of the T-tubules that form the

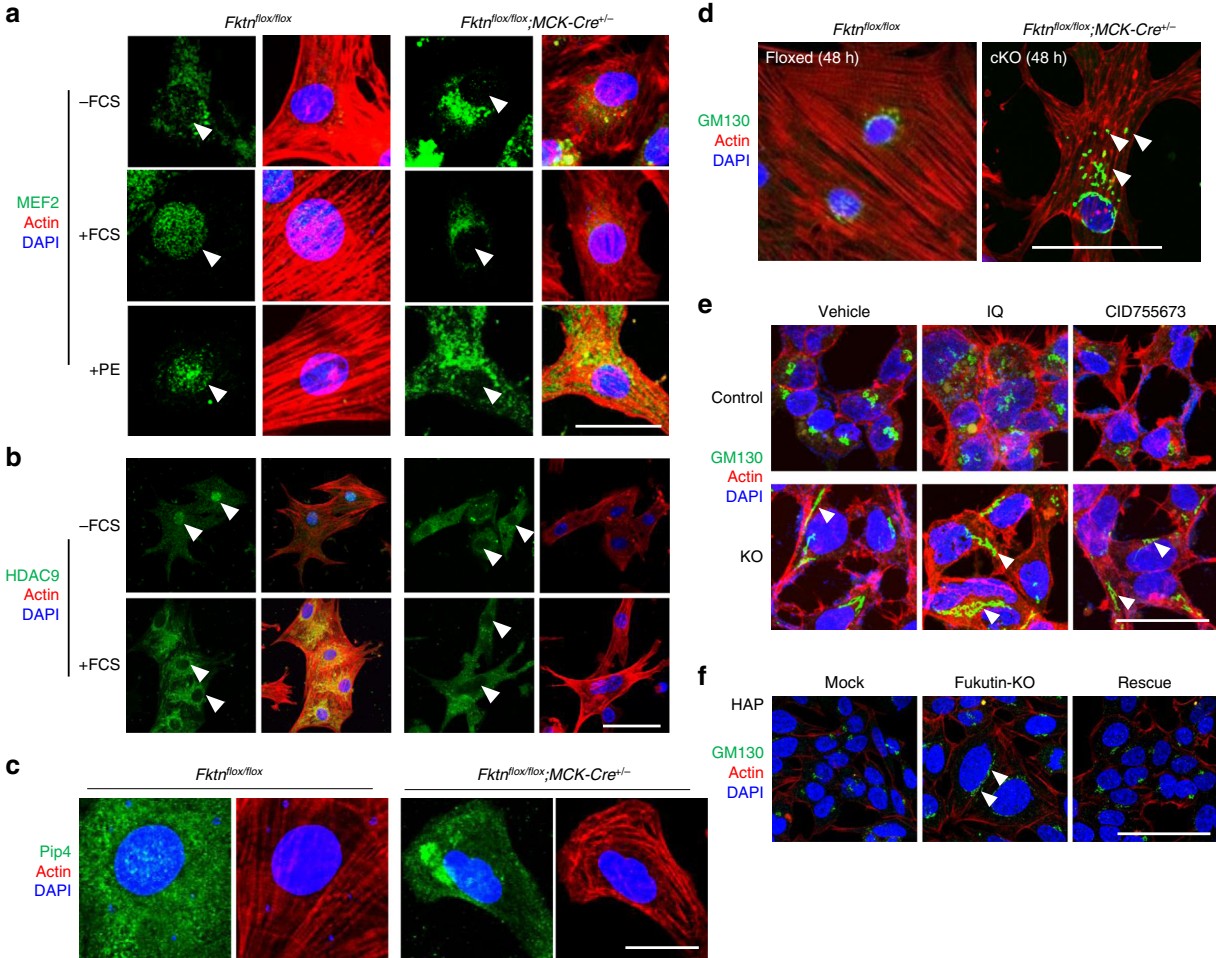

**Fig. 5 Impaired MEF2-HDAC axis of MCK-*Fktn*-cKO (*Fktn*$^{flox/flox}$; *MCK-Cre*$^{+/-}$) cardiomyocytes. a** Representative images of MEF2 localization in cardiomyocytes treated with FCS or PE for 48 h. Scale bar, 20 μm. **b** Representative images of HDAC9 localization in cardiomyocytes cultured with or without FCS. Scale bar, 100 μm. **c** Representative images of PI4P accumulation in cardiomyocytes. Scale bar, 20 μm. **d** Representative images of GM130 distribution in cardiomyocytes. Scale bar, 50 μm. **e** Effects of a PKD activator (ilimaquinone (IQ)) and a PKD inhibitor (CID755673) on GM130 distribution in control and *FKTN*-deficient (KO) HEK cells. Arrowheads show abnormally distributed GM130 signals. Scale bar, 100 μm. **f** Representative images of GM130 distribution in control (mock), *FKTN*-deficient (fukutin-KO), and rescued (fukutin-KO transfected with pcDNA-FKTN-FLAG) HAP cells. Arrowheads show abnormally distributed GM130 signals. Scale bar, 100 μm.

key structure of E–C coupling, independent of α-DG glycosylation. Despite the severe structural disorganization of T-tubules, the expression of Ca$^{2+}$ regulatory proteins, including NCX1 and LTCC, was not altered (Fig. 7c, d and Supplementary Fig. 9).

To investigate the mechanism of T-tubule disorganization in *Fktn*-deficient cardiomyocytes, we analyzed the expression levels of junctophilin-2 (JP2) (Fig. 7c, e), a protein that bridges T-tubules and SR membranes[34,35]. It is well known that misregulation of JP2 can be mediated via microtubules (MTs) and contributes to T-tubule maintenance and Ca$^{2+}$ mishandling in failing myocytes[34,35]; therefore, we also analyzed the expression of α-tubulin, a component of MTs, in hearts from tamoxifen-treated αMHC-MCM-*Fktn*-cKO mice (Fig. 7c, f). JP2 expression was clearly reduced in the hearts of tamoxifen-treated αMHC-MCM-*Fktn*-cKO mice (Fig. 7c, e), whereas α-tubulin expression was significantly increased (Fig. 7c, f). Moreover, the distribution of JP2 was altered in tamoxifen-treated αMHC-MCM-*Fktn*-cKO cardiomyocytes (Fig. 7g). Therefore, we hypothesized that MT densification-mediated JP2 redistribution may be associated with the cardiomyocyte contractile defects via T-tubule remodeling in tamoxifen-treated αMHC-MCM-*Fktn*-cKO mice.

**The effects of MT depolymerization in αMHC-MCM-*Fktn*-cKO myocytes**. Because MT densification can impede myocyte shortening[35–38], we examined whether treatment with 10 μM colchicine, which inhibits MT polymerization, for 1 h resolved contractile dysfunction in tamoxifen-treated αMHC-MCM-*Fktn*-cKO cardiomyocytes. In control cardiomyocytes, α-tubulin immunostaining revealed a regulated meshwork pattern of MTs (Fig. 8a, top left), with some polymerized MTs perpendicular to the muscle fiber (Fig. 8a, top left inset). The meshwork pattern of MTs was lost after treatment with 10 μM colchicine (Fig. 8a, middle left), whereas cardiomyocyte shortening was not affected in vehicle-treated control myocytes (Fig. 8b). In tamoxifen-treated αMHC-MCM-*Fktn*-cKO cardiomyocytes, we observed an accumulation of MTs in parallel with the muscle fiber (Fig. 8a, top right and inset), and shortening was impaired compared with that in the vehicle control (Fig. 8b). Notably, colchicine treatment reduced this accumulation of MTs and reversed the impaired shortening in tamoxifen-treated αMHC-MCM-*Fktn*-cKO cardiomyocytes (Fig. 8a, bottom right; Fig. 8b). Golgi elements in muscles are major sites for the nucleation of MTs[39], and the irregular accumulation of MTs, due to the disorganization and

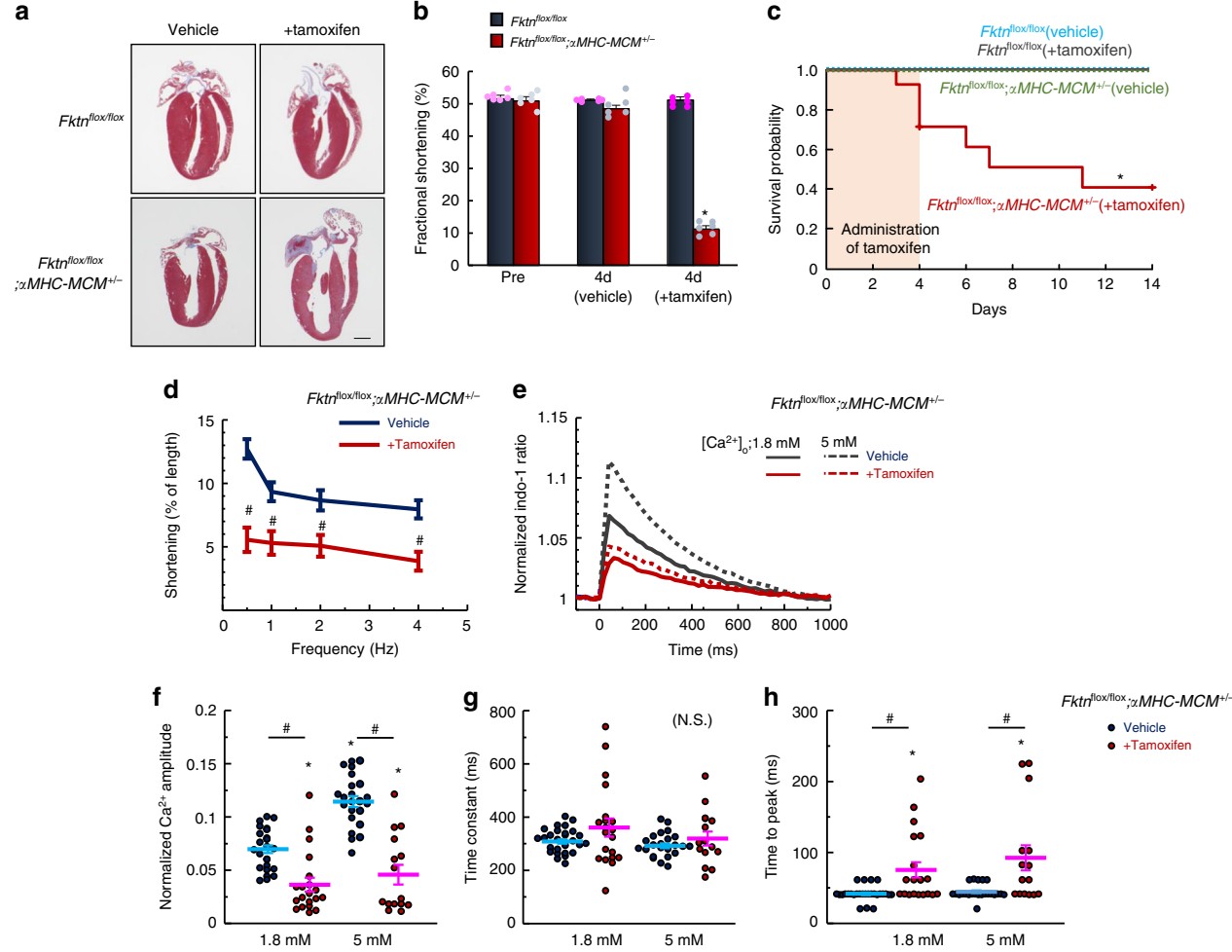

**Fig. 6 Acute elimination of cardiac *Fktn* from 10-week-old mice.** Cardiac morphology (scale bar, 1 mm) (**a**) and function ($n = 6$ per group) (**b**) in control (floxed) and αMHC-MCM-*Fktn*-cKO (*Fktn*$^{flox/flox}$; αMHC-MCM$^{+/-}$) mice after 4 d of tamoxifen treatment. *$P < 0.05$ vs. floxed mice before tamoxifen treatment based on Dunnett's post-hoc tests. **c** Survival probabilities ($n = 14$ per group). *$P < 0.05$ vs. vehicle-treated floxed mice based on log-rank tests. **d** Frequency-dependent shortening of myocytes isolated from mice treated with or without tamoxifen for 4 d ($n = 21–30$ cells measured from 3 hearts per group). #$P < 0.05$ vs. vehicle-treated mice based on Student's t-tests. **e** Indo-1 fluorescence in single cardiomyocytes stimulated at 1 Hz (1.8 mM Ca$^{2+}$, vehicle, $n = 27$ cells; +Tamoxifen, $n = 20$ isolated from 3 hearts per groups; 5 mM Ca$^2$, vehicle, $n = 23$ cells; +Tamoxifen, $n = 15$ isolated from 3 hearts per group). Normalized peak amplitude (**f**), decay time constant (obtained by fitting the decline phase) (**g**), and time to peak (**h**) of Ca$^{2+}$ transients. *$P < 0.05$ vs. myocytes from floxed mice at 1.8 mM Ca$^{2+}$; #$P < 0.05$ between indicated groups based on Tukey–Kramer tests.

abnormal distribution of the Golgi apparatus, has recently been implicated in the progression of heart failure[34,35,40,41]. Interestingly, treatment with 10 μM colchicine disrupted the Golgi apparatus (Fig. 8a, middle left), consistent with the idea that MTs are crucial for proper localization of the Golgi in a wide range of cells[34]. Remarkably, this fragmentation of the Golgi did not affect cardiomyocyte shortening (Fig. 8b). Instead, the activation of PKD with ilimaquinone resulted in vesiculation in the Golgi apparatus, an accumulation of MTs (Fig. 8a, bottom left and inset), and contractile dysfunction (Fig. 8b) in control cardiomyocytes. In tamoxifen-treated αMHC-MCM-*Fktn*-cKO cardiomyocytes, we also observed abnormal fragmentation of the Golgi apparatus (Fig. 8a, top right, white arrowheads). These results suggest that contractile dysfunction in FKTN-deficient cardiomyocytes is caused by MT densification that may be associated with Golgi abnormalities. In tamoxifen-treated αMHC-MCM-*Fktn*-cKO hearts, the level of PKD phosphorylation was significantly higher than that in control hearts (Fig. 8c), suggesting that FKTN impacts the integrity of the Golgi structure by regulating PKD signaling pathways. We also observed abnormal distribution of the Golgi apparatus and the accumulation of MT

structures in MCK-*Fktn*-cKO mice in late adulthood (Supplementary Fig. 10). Control myocytes from 10-month-old mice also exhibit the accumulation of MT structures. While membrane abnormalities were observed in cardiomyocytes isolated from 10-month-old MCK-*Fktn*-cKO mice (Supplementary Fig. 10), the severity was less than that observed in tamoxifen-treated αMHC-MCM-*Fktn*-cKO hearts (Fig. 7a). Additionally, phosphorylation of PKD was decreased in cardiomyocytes isolated from MCK-*Fktn*-cKO mice 48 weeks after birth (Supplementary Fig. 11). On the other hand, reduction of FKTN protein immediately led to the abnormal fragmentation of the Golgi apparatus, MT densification, and PKD phosphorylation in tamoxifen-treated αMHC-MCM-*Fktn*-cKO hearts (Fig. 8). These data suggest that FKTN plays a crucial role in the maintenance of the Golgi-derived MT network and PKD signaling.

To elucidate how colchicine improves myocyte contractility, we assessed Ca$^{2+}$ handling during E–C coupling in cardiomyocytes isolated from tamoxifen-treated αMHC-MCM-*Fktn*-cKO hearts. We found that colchicine treatment enhanced intracellular Ca$^{2+}$ handling in tamoxifen-treated αMHC-MCM-*Fktn*-cKO myocytes (Fig. 8d). This was associated with improved myocyte contractile

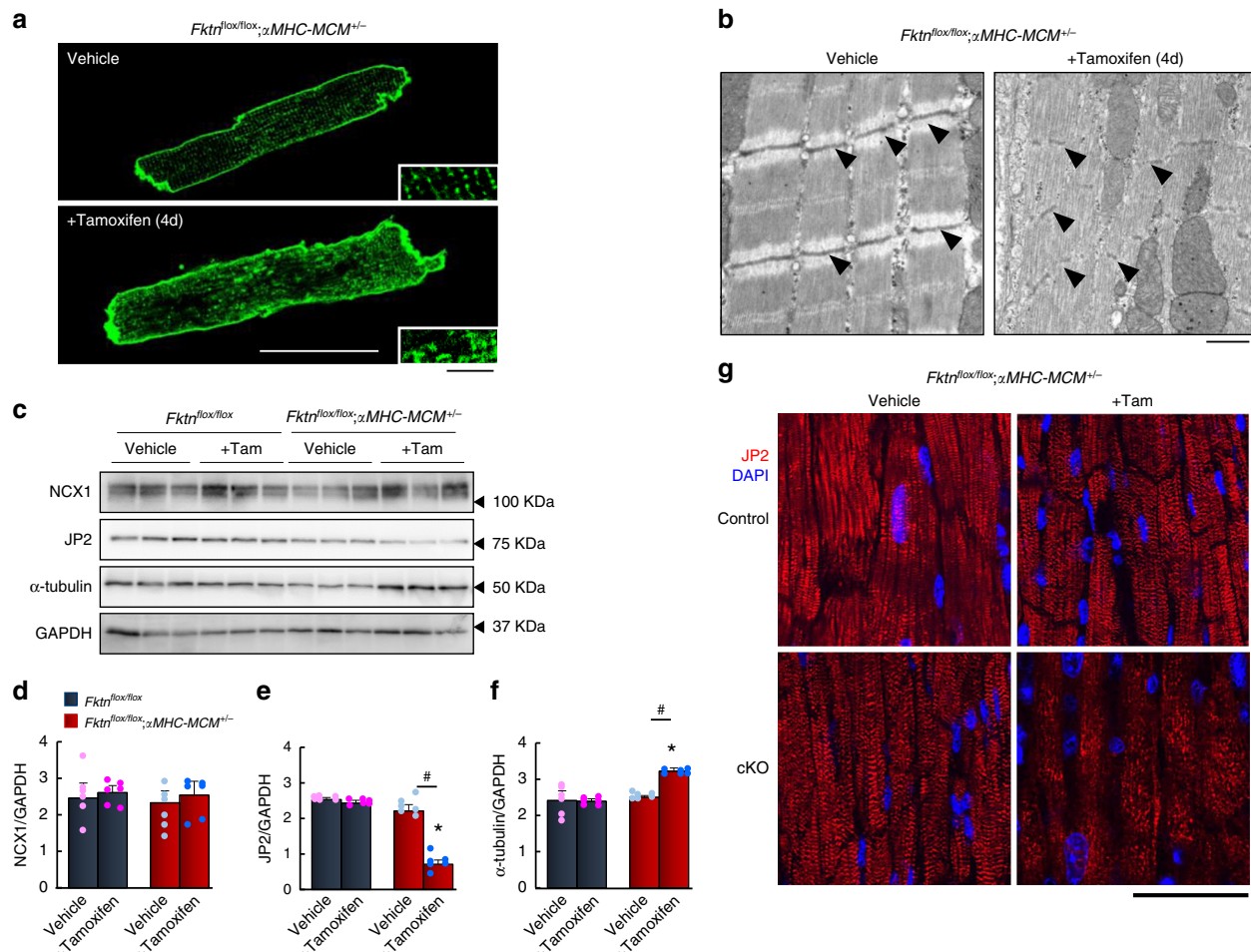

**Fig. 7 Subcellular change in αMHC-MCM-*Fktn*-cKO (*Fktn*[flox/flox]; *αMHC-MCM*[+/−]) myocytes. a** Representative membrane staining by di-8-ANEPPS. Scale bar, 50 μm (inset, 10 μm). **b** Representative electron micrographs of myofilaments in vehicle- and tamoxifen-treated αMHC-MCM-*Fktn*-cKO mice. Arrowheads show Z-lines. Scale bar, 1 μm. Representative immunoblots (**c**) and expression levels (**d–f**) for NCX1, JP2, and α-tubulin in floxed (control) and αMHC-MCM-*Fktn*-cKO mice with or without tamoxifen. GAPDH was used as a loading control (*n* = 6 mice per group). *$P < 0.05$ vs. vehicle-treated floxed mice; #$P < 0.05$ between indicated groups based on Tukey–Kramer tests. **g** Representative immunostaining for JP2 (red) and DAPI (blue). Scale bar, 50 μm.

efficiency as evidenced by the ratio of cell shortening to the amplitude of $Ca^{2+}$ increase during E–C coupling (Fig. 8e). Thus, colchicine appears to improve myocyte contractility by improving $Ca^{2+}$ handling and reducing the stiffness of cardiomyocytes (Fig. 8e).

Finally, we showed that colchicine ameliorated the heart failure phenotype in tamoxifen-treated αMHC-MCM-*Fktn*-cKO mice. Chamber dilation, fibrosis, cardiac dysfunction, and survival rates after 10 d of tamoxifen treatment were all improved by colchicine administration (Fig. 9a–e). These observations suggest acute elimination of *Fktn* contributes to myocyte contractile dysfunction as a result of MT densification.

**Mechanistic analysis for Golgi pathology in *Fktn*-deficient hearts.** To elucidate how FKTN regulates Golgi structure and function and thus leads to MT accumulation, we performed microarray analysis of tamoxifen-treated floxed or αMHC-MCM-*Fktn*-cKO hearts and investigated the putative pathway and biological processes induced by deletion of *Fktn*. A total of 31,762 probe sets obtained from an Agilent gene expression microarray platform (SurePrint G3 Mouse Gene Expression 8 × 60 K) were tested for differential expression. We found that 7408 genes were differentially expressed on the basis of a criterion of at least a twofold

change and adjusted value of $P < 0.05$ when cKO was compared with floxed-control; 4536 transcripts were downregulated and 2872 transcripts were upregulated in αMHC-MCM-*Fktn*-cKO hearts (Table I in the online-only Data Supplement). Hierarchical clustering revealed a high level of homogeneity within and a clear separation between groups (Supplementary Fig. 12). A heatmap displaying expression levels of 1,611 differentially expressed Golgi apparatus-related genes annotated from a global cross-database NCBI search showed excellent consistency of changes within groups (Fig. 10a). Thus, the elimination of *Fktn* impacts the expression profile of Golgi-related genes.

Integrity of the Golgi apparatus requires the MT network[42]. We focused on genes known to be relevant to nucleation and dynamics of Golgi-derived MTs: *Tpx2* (coding for a targeting protein of Xklp2), *Rab3a* (coding for RAS oncogene family member 3a), *Pde4dip* (coding for myomegalin), *Pcnt* (coding for pericentrin), and *Clasp1* (coding for CLIP-associating protein 1)[42]. The positions of these genes in the volcano plot are depicted in Fig. 10b. Although the MT nucleation factor encoded by *Tpx2* was increased, the anchoring proteins of the γ-tubulin ring complex (γ-TuRC) of the *cis*-Golgi encoded by *Pde4dip* and *Pcnt* were reduced in cKO hearts. The MT-stabilizing protein encoded by *Clasp1* was also reduced in cKO hearts. These observations suggest that *Fktn* elimination leads to a decline in the stability and dynamics of the Golgi-derived

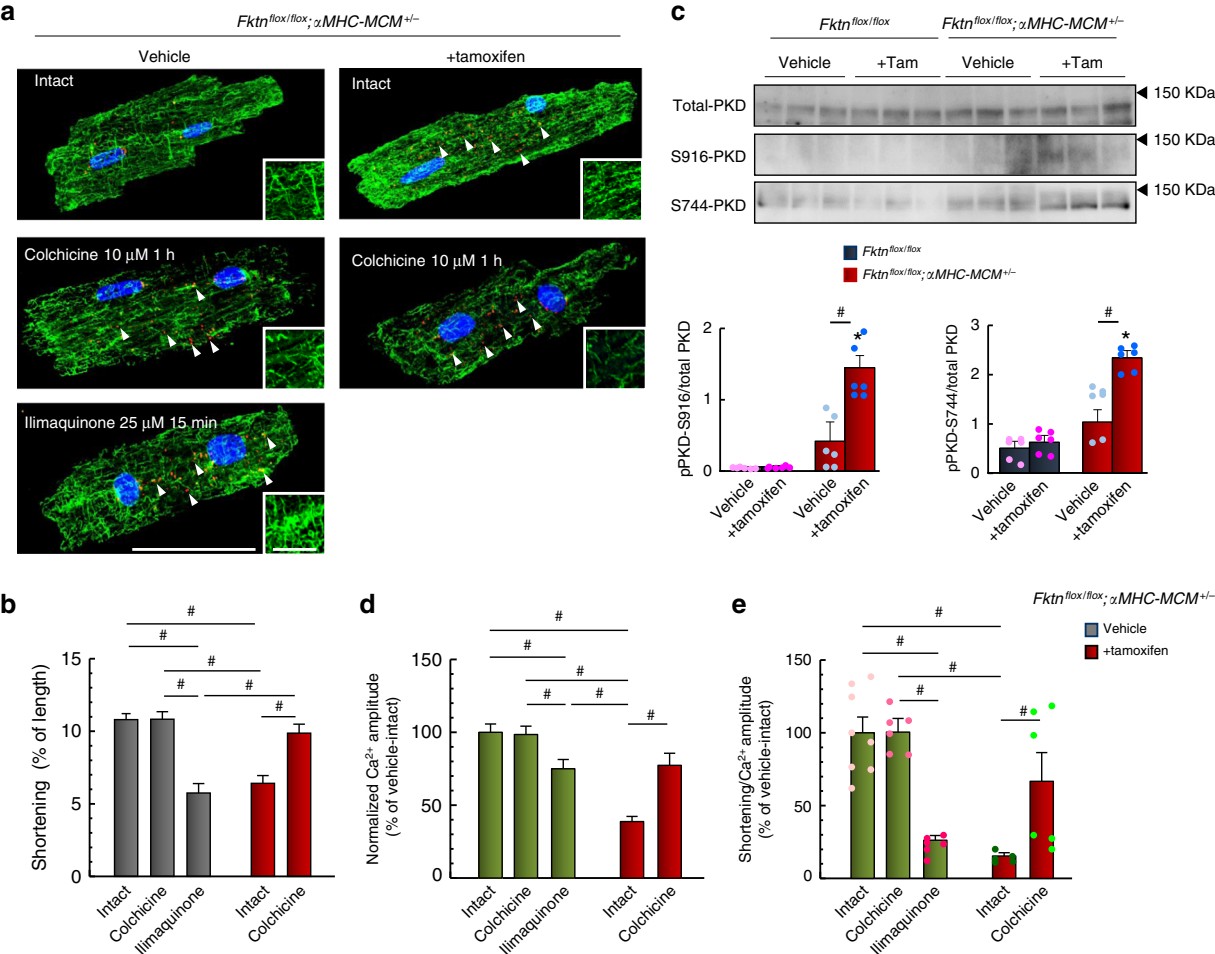

**Fig. 8 MT densification and fragmentation of Golgi in αMHC-MCM-*Fktn*-cKO (*Fktn*^flox/flox^; *αMHC-MCM*^+/−^) myocytes. a** Representative immunofluorescence images for MT (green) and GM130 (red) staining. Arrowheads show abnormally distributed GM130 signals. Scale bar, 50 μm (insets, 10 μm). **b** Electrical stimulation-induced (1 Hz) myocyte shortening (vehicle-intact, $n = 101$ cells from 5 mice; vehicle-colchicine, $n = 40$ cells from 4 mice; vehicle-ilimaquinone, $n = 20$ cells from 3 mice, tamoxifen-intact, $n = 29$ cells from 3 mice; tamoxifen-colchicine, $n = 40$ cells from 3 mice). **c** Representative images of immunoblots. Phosphorylation level of PKD is shown relative to total PKD ($n = 6$ mice per group). *$P < 0.05$ vs. vehicle-treated floxed mice; #$P < 0.05$ between indicated groups based on Tukey–Kramer tests. **d** Normalized peak amplitudes (vehicle-intact, $n = 32$ cells from 4 mice; vehicle-colchicine, $n = 39$ cells from 3 mice; vehicle-ilimaquinone, $n = 26$ cells from 3 mice, tamoxifen-intact, $n = 14$ cells from 4 mice; tamoxifen-colchicine, $n = 11$ cells from 4 mice). **e** Contractile efficiency (vehicle-intact, $n = 8$ cells; vehicle-colchicine, $n = 6$ cells; vehicle-ilimaquinone, $n = 6$ cells, tamoxifen-intact, $n = 5$ cells; tamoxifen-colchicine, $n = 6$ cells from 3 mice per group). #$P < 0.05$ between indicated groups based on Tukey–Kramer tests (**b**, **d**, **e**).

MT network. In addition, the downregulation of *Rab3a* suggests a decline in post-Golgi secretory trafficking[43] in cKO hearts.

Pathway enrichment analysis of the downregulated genes in cKO hearts (FDR < 0.001, Log$_2$FC < −1.5) revealed enrichment of pathways involved in cardiac physiology (i.e., muscle contraction and cardiac conduction; Supplementary Fig. 12). On the other hand, upregulated genes in cKO hearts (FDR < 0.001, Log$_2$FC < −1.5) revealed enrichment of pathways involved in the cell cycle, Rho GTPase signaling, extracellular matrix organization or degradation, integrin cell surface interaction, and extracellular matrix proteoglycan as analyzed by ReactomePA (https://reactome.org/; Fig. 10c). These pathways were influenced by the stability and dynamics of Golgi-derived MT, protein synthesis at the Golgi apparatus, and membrane transport via MT in cells. Thus, these observations emphasize the possibility that disintegration of the Golgi apparatus and Golgi-derived MT network is one etiology of muscular dystrophy-associated heart failure in *Fktn*-deficient mice.

Kyoto Encyclopedia of Gene and Genomes (KEGG) functional network analysis of the 7408 genes showed large fluctuations with high significance for Parkinson's, Huntington's, and Alzheimer's diseases (Fig. 10d). Structural and functional alteration of the Golgi apparatus is recognized as a constant pathological hallmark of neurodegenerative disease, including Parkinson's, Huntington's, and Alzheimer's diseases[43]. A pathway related to hypertrophic cardiomyopathy and dilated cardiomyopathy was also prominent. These results suggest that the pathological mechanism of cardiac dysfunction induced by *Fktn* elimination partly overlaps with that of neurodegenerative disease.

To investigate the cause of the Golgi pathology observed in *Fktn* elimination, we examined localization of the α-DG protein using antibody against the core region of α-DG (core α-DG) in cultured neonatal cardiomyocytes isolated from MCK-*Fktn*-cKO hearts (Fig. 10e). In control myocytes, core α-DG co-localized with the post-Golgi marker Rab3a beneath the membrane (Fig. 10e, upper panels). On the other hand, although core

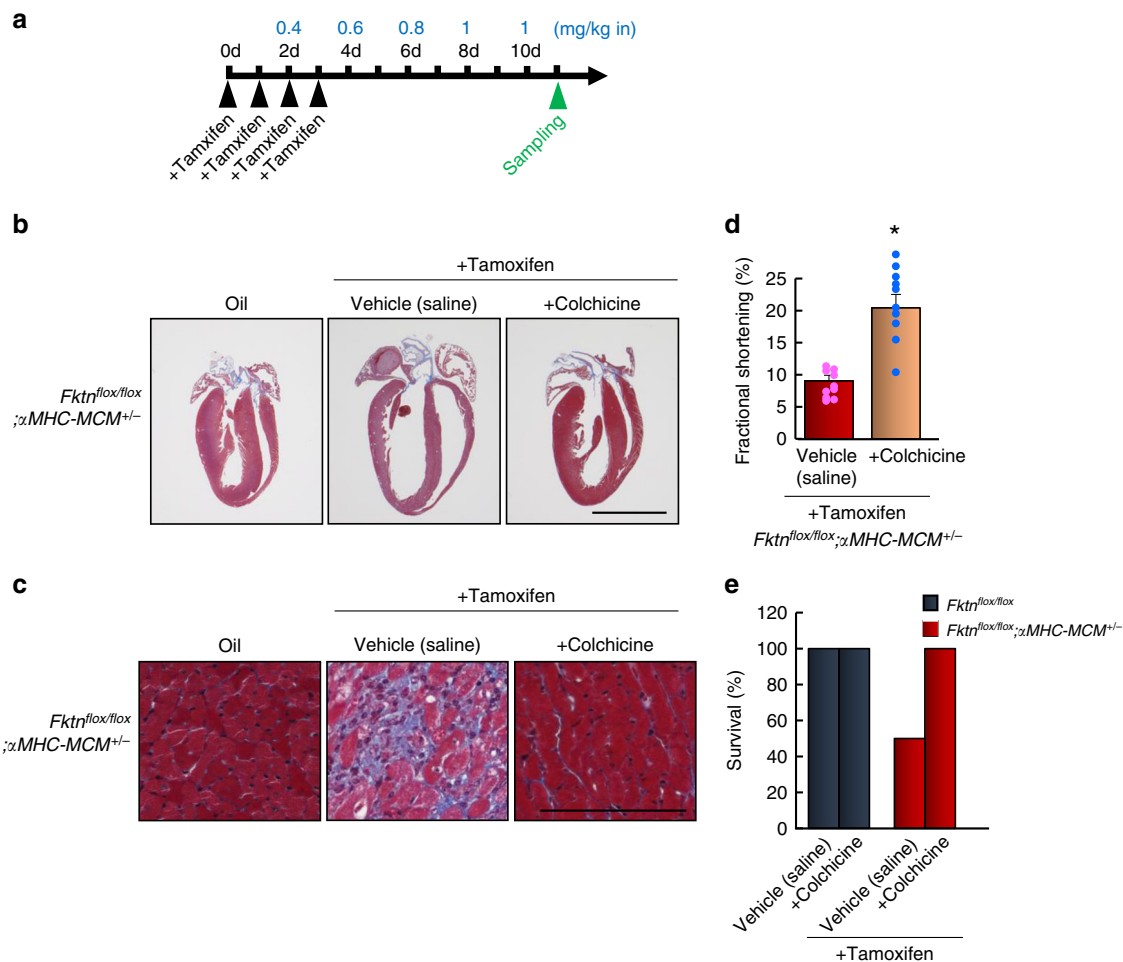

**Fig. 9 The effects of MT depolymerization in αMHC-MCM-*Fktn*-cKO (*Fktn*$^{flox/flox}$; *αMHC-MCM*$^{+/−}$) hearts. a** Schedule of tamoxifen administration, colchicine treatment, and sample collection. Blue letters show the dose of colchicine. **b** Cardiac morphology (scale bar, 5 mm). **c** Masson's trichrome staining of the left ventricle (scale bar, 50 μm). Cardiac function in tamoxifen-treated αMHC-MCM-*Fktn*-cKO hearts treated with colchicine (**d**) and survival percentages (**e**) (n = 10 per group). *P < 0.05 vs. tamoxifen-treated αMHC-MCM-*Fktn*-cKO mice based on Student's t-tests.

α-DG also co-localized with Rab3a in cKO myocytes, its signal accumulated intracellularly on the vesicle structure (Fig. 10e, lower panels). Next, we examined whether accumulation of core α-DG is a critical factor in disorganization of the Golgi apparatus (Fig. 10f). We observed faint signals of anti-core α-DG antibody beneath the membrane in HEK cells (Fig. 10f, upper panels). On the other hand, *FKTN*-KO HEK cells showed internally accumulated core α-DG, which mainly co-localized with Rab3a, and fragmentation of the Golgi apparatus (Fig. 10f, middle panels). Forced expression of full-length DG induced accumulation of core α-DG in the post-Golgi area, which co-localized with Rab3a (Fig. 10f, left lower panels). However, we did not observe fragmentation of the Golgi apparatus by GM130 labeling (Fig. 10f, right lower panels). These observations suggest that accumulation of core α-DG does not trigger the Golgi pathology observed in *Fktn*-deficient cells and thus that FKTN may be critical for the maintenance of Golgi structure and function.

## Discussion

As DG glycosylation and the DGC play essential roles in the connection between the basement membrane and sarcolemma in skeletal muscle[23,44], the loss of membrane integrity has been thought to be a cause of muscle damage in cardiac tissues[45]. In support of this hypothesis, it was shown that DG limits cardiac myocyte membrane damage[46]. In addition, our present study

demonstrates, for the first time, the cellular and molecular pathomechanisms of heart failure associated with α-DGpathy.

Studies of skeletal muscle-specific *Fktn*-cKO mice indicate that the loss of functional α-DG protein likely contributes to the pathogenesis of muscular dystrophy[23,44]. Surprisingly, our study shows a lack of cardiac dysfunction in young-adult MCK-*Fktn*-cKO mice despite the marked reduction in DGC proteins in the cardiac sarcolemma, suggesting that membrane fragility is not the sole etiology of cardiac dysfunction. Cardiac dysfunction with impaired myocyte shortening in old age (Figs. 1 and 2) or vulnerability to hemodynamic stress at a young age (Fig. 3) in MCK promoter-driven cKO hearts suggests that the FKTN protein is critical for maintaining cardiac structure and function under conditions of aging and hemodynamic stress. Therefore, MCK promoter-driven cKO hearts may have an unknown compensatory mechanism that prevents mice from manifesting any obvious phenotypes of FKTN deficiency during the young-adult age under normal physiological conditions. Therefore, it is possible that the acute *Fktn* deficiency results in severe cardiac dysfunction and reduced survival in the young mice. Notably, DGC localization was not affected in these young mice, suggesting that reduced α-DG glycosylation and DGC expression may not be the sole reasons for cardiac dysfunction.

A main finding is that FKTN is crucial for maintaining contractile function and Ca$^{2+}$ handling during E–C coupling in

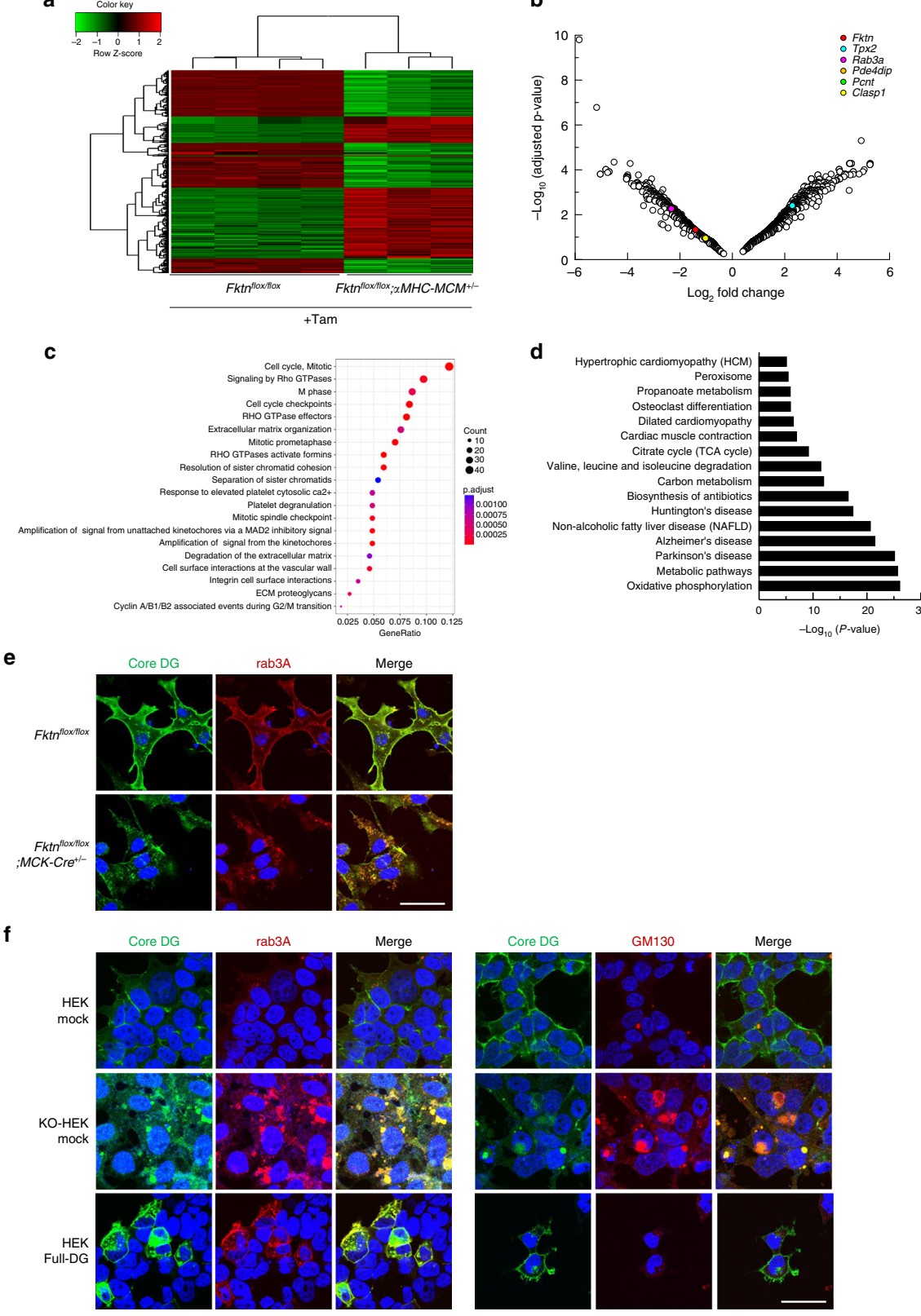

individual cardiomyocytes. By contrast, previous studies of dystrophin-deficient *mdx* mice found changes in $Ca^{2+}$ signaling with only slightly elevated resting intracellular concentrations and potentially arrhythmogenic E–C coupling in ventricular myocytes[47,48]. Homeostasis in young-adult *Fktn*-deficient mice may have been achieved by reduced $Ca^{2+}$ influx through LTCC and increased efflux by NCX1 or SERCA (Supplementary Fig. 13), proteins that play important roles in $Ca^{2+}$ homeostasis and $Ca^{2+}$ handling during E–C coupling[27,49,50]. These alterations indicate that reserved capacity in responses to cardiomyocytes stress was impaired without causing significant cardiac dysfunction. Chronic isoproterenol treatment in these mice downregulated

**Fig. 10 Fktn elimination impacts Golgi-related genes and structure. a** Microarray analysis using floxed and αMHC-MCM-Fktn-cKO (Fktn^flox/flox; αMHC-MCM^+/−) hearts treated with tamoxifen for 4 d. Heat map displaying hierarchical clustering and expression levels of differentially expressed Golgi apparatus-related genes. **b** Volcano plot of upregulated or downregulated genes in tamoxifen-treated αMHC-MCM-Fktn-cKO (Fktn^flox/flox; αMHC-MCM^+/−) hearts. Dots indicating Pde4dip, Pcnt, and Clasp are overlapping. **c** Gene set enrichment analysis revealing enriched pathways and processes in tamoxifen-treated αMHC-MCM-Fktn-cKO hearts. **d** Significantly enriched Kyoto Encyclopedia of Genes and Genomes (KEGG) categories show differentially expressed gene pathways in tamoxifen-treated αMHC-MCM-Fktn-cKO hearts. **e** Representative immunofluorescence images of neonatal cardiomyocytes cultured with FCS (Core DG, green; Rab3a, red; DAPI, blue). Scale bar, 50 μm. **f** Representative immunofluorescence images of HEK, FKTN-KO HEK, and HEK expressing full-length DG (Core DG, green; Rab3a, red; DAPI, blue (left panels); Core DG, green; GM130, red; DAPI, blue (right panels)). Scale bar, 50 μm.

Akt and upregulated CamKII, suggesting altered $Ca^{2+}$-dependent signaling. Thus, the inability to precisely regulate intracellular $Ca^{2+}$ after Fktn elimination likely impacted cell survival in the long run. Indeed, we also observed an upregulation of the hypertrophic mediator calcineurin[51] and a downregulation of its endogenous inhibitor MCIP[52], along with strong induction of the hypertrophic marker ANP, in MCK-Fktn-cKO mice at 16 weeks of age (Supplementary Fig. 13). Thus, at this age, hypertrophic remodeling, mediated by $Ca^{2+}$-calcineurin signaling, is already underway before abnormalities in morphology and contractility emerge. In our study, Fktn-deficient cardiomyocytes showed hyperphosphorylation and altered localization of HDAC9, which is a direct transcriptional target of MEF2 and acts as signal-responsive repressor of cardiac hypertrophy[31]. HDAC9-knockout mice are ultrasensitive to hypertrophic stimuli and spontaneously develop cardiac hypertrophy with advanced age[31]. Thus, HDAC9 alterations can be associated with the vulnerability to stress in Fktn-deficient myocytes. The constitutive phosphorylation of PKD, which transduces signals for diacylglycerol biogenesis[53], and hyperphosphorylation of HDAC9 under basal conditions in the hearts of Fktn-deficient mice suggest there may be excessive transcriptional activation in response to stress.

Another key finding is that FKTN is essential for maintaining the Golgi-derived MT network in cardiomyocytes. Constitutively phosphorylated PKD in myocytes from young MCK-Fktn-cKO mice or tamoxifen-treated αMHC-MCM-Fktn-cKO myocytes may remain associated with the cytoplasmic surfaces of Golgi membranes and regulate the fission of vesicles that carry protein and lipid cargo to the plasma membrane[54]. We also observed the vesiculation of Golgi and accumulation of MT in the myocytes from old MCK-Fktn-cKO mice (Supplementary Fig. 10), supporting the notion that FKTN is involved in the maintenance of the Golgi-MT network. Alternatively, PKD phosphorylation in myocytes from old MCK-Fktn-cKO mice was lower than that in control myocytes (Supplementary Fig. 11). This contradiction may be explained by differences in timing of Fktn deletion, disease progression course (period), and hemodynamic stress conditions, thus reflecting pathogenesis and/or pathological consequences. Nonetheless, our data show that FKTN protein plays a role in the maintenance of Golgi-MT network in cardiomyocytes. Considering the results that old MCK-Fktn-cKO mice showed very mild pathology compared to isoproterenol-treated young MCK-Fktn-cKO mice or tamoxifen-treated αMHC-MCM-Fktn-cKO mice, our data also suggest the presence of a compensatory mechanism during long-term progression in the presence of Fktn deficiency.

The altered distribution of GM130 in FKTN-deficient HAP and HEK cells indicates that the Golgi morphology was also altered independently of PKD activity (Fig. 5e, f). Impairment of glycosylation by Fktn deficiency may accelerate fragmentation of the Golgi apparatus by enhancing the budding of vesicles from the Golgi membrane or disrupting protein trafficking. We observed the accumulation of core α-DG in the post-Golgi area and fragmentation of the Golgi apparatus in Fktn-deficient

cardiomyocytes and HEK cells (Fig. 10e, f). HEK cells expressing full-length α-DG also showed accumulation of core α-DG in the post-Golgi area (Fig. 10f). However, we did not observe abnormal fragmentation of the Golgi apparatus in cells, suggesting a pivotal role of FKTN in the maintenance of Golgi integrity (Fig. 10f). FKTN is targeted to the medial Golgi apparatus through its N-termini and transmembrane domain[55]. Therefore, the lack of FKTN per se might affect the integrity of the Golgi ribbon. Alternatively, FKTN also forms a complex with other Golgi-resident glycosyltransferases[56,57], and thus, the lack of FKTN may change cellular glycosylation, affecting Golgi structure and function. The results of our microarray analysis also suggest the impact of FKTN on expression of Golgi-related genes. Elimination of Fktn led to the downregulation of genes encoding the anchoring protein of γ-TuRC at cis-Golgi and MT-stabilizing protein (Fig. 10b). Golgi-derived MT networks serve important roles in post-Golgi trafficking, maintenance of Golgi integrity, cell polarity, and cellular function[42]. The disruption of MT dynamics in Fktn-deficient myocytes may impact Golgi vesiculation, with the accumulation and densification of MTs affecting both the central cellular localization of Golgi and ER–Golgi–plasma membrane trafficking, which indirectly impacts the size and morphology of the Golgi[58]. Recent studies show that disintegration of Golgi structure and function is one of the main causes of Alzheimer's disease[42]. In this disease, precipitation of hyperphosphorylated tau protein is also related to Golgi structure and function[42]. Remarkably, KEGG analysis suggested that heart failure induced by acute Fktn elimination has a common etiology with neuromuscular disease (Fig. 10d). Our results further suggest a mechanistic connection between the Golgi apparatus and myocyte T-tubule organization, which is crucial for $Ca^{2+}$ homeostasis in cardiomyocytes. Further investigations will clarify the FKTN-dependent maintenance of Golgi-derived MT networks.

Cardiomyocytes isolated from acute Fktn deficiency in young mice had dramatically disrupted T-tubule membranes and MT densification, as seen in failing hearts of human patients and animal models[35,59,60]. Studies show that MT-mediated defects in JP2 trafficking contribute to myocyte T-tubule remodeling in heart failure[34,38]. Here, we show that Fktn elimination also led to a reduction and redistribution of JP2. Thus, disorganization of the T-tubule membrane in Fktn-deficient mice might be caused by MT-mediated defects in JP2 trafficking. These changes contributed to reduced myocyte contractility and cardiac dysfunction, which were ameliorated by colchicine treatment. These data suggest that the accumulation of MTs after Fktn elimination is a major contributor to myocyte dysfunction and that an effective therapeutic strategy for heart failure involves MT depolymerization. MT densification increases the stiffness of cells, thereby mechanically impeding sarcomere motion[40], and alters T-tubules, which are important for E–C coupling[34]. We show that treatment with colchicine not only restored myocyte shortening, $Ca^{2+}$ handling during E–C coupling, and contractile efficiency but also ameliorated severe cardiac dysfunction and improved the survival

of mice with FKTN deficiency. One question is whether colchicine treatment can be applied to other forms of muscular dystrophy-associated cardiomyopathies. Answering this question requires a system that can tightly control the timing of eliminating the expression of the responsible gene, as we did in this study. However, a previous report suggested that colchicine treatment improves $Ca^{2+}$ handling in $mdx$ mice[38], a model for DMD; although, the exact mechanism remains to be elucidated. Importantly, as colchicine is already an FDA-approved drug for the treatment of familial Mediterranean fever and acute gout flares, our findings suggest that this drug may also be useful for treating muscular dystrophy-associated heart failure.

Several studies have raised concerns that tamoxifen treatment may adversely influence cardiac function in adult MCM mice[61,62]. DNA damage response occurs in these hearts, resulting in cardiomyocyte apoptosis, cardiac fibrosis, and cardiac dysfunction[61]. Therefore, Koitabashi et al.[63] recommended a reduced tamoxifen dosage of <20 mg/kg of body weight per day for use in $\alpha$MHC-MCM[+/−] mice (an approximate total dose of 80 mg/kg). In this study, we administered tamoxifen by intraperitoneal (i.p.) injection consecutively for 4 d at a dosage of 8 mg/kg of body weight in 10-week-old $Fktn^{flox/flox}$;MCM[+/−], $Fktn^{flox/+}$;MCM[+/−], and $Fktn^{flox/flox}$;MCM[−/−] mice. This dose (8 mg/kg per d, approximate total dose of 32 mg/kg) was sufficient for reducing FKTN protein expression and $Fktn$ transcription (Supplementary Fig. 6e and Fig. 10b). In our microarray analysis, we were unable to detect any alterations in the signaling pathways involved in DNA damage response in $Fktn^{flox/flox}$; $\alpha$MHC-MCM[+/−] hearts treated with tamoxifen for 4 d (Fig. 10). In addition, tamoxifen treatment had no effect on the overall cardiac structure and function of $Fktn^{flox/+}$;$\alpha$MHC-MCM[+/−] (hetero) mice (Supplementary Fig. 7). Together, our results show that the heart-failure phenotype seen in tamoxifen-treated $\alpha$MHC-MCM-$Fktn$-cKO ($Fktn^{flox/flox}$; $\alpha$MHC-MCM[+/−]) mice is not due to adverse effects of tamoxifen or excessive MCM expression. After 4 d of tamoxifen treatments, $Fktn$ mRNA levels were reduced by about 80% in the hearts of $\alpha$MHC-MCM-$Fktn$-cKO ($Fktn^{flox/flox}$;$\alpha$MHC-MCM[+/−]) mice (Supplementary Fig. 7c). In addition, FKTN protein expression was reduced by 80–90% in $\alpha$MHC-MCM-$Fktn$-cKO ($Fktn^{flox/flox}$;$\alpha$MHC-MCM[+/−]) mice after 4 d of tamoxifen treatment or 6 d after the onset of tamoxifen treatment compared with vehicle control hearts (Supplementary Fig. 6e). These data also suggest that the half-life of FKTN protein is <4 d. Thus, FKTN mRNA and/or FKTN protein likely have a fast turnover rate. Altogether, our data strongly suggest that acute reduction of FKTN protein expression results in severe cardiac dysfunction in tamoxifen-treated $Fktn^{flox/flox}$; $\alpha$MHC-MCM[+/−] mice. Thus, FKTN protein is crucial for the maintenance of cardiac structure and function in mice.

## Methods

**Animals**. All the mouse experiments have been done according to the all relevant ethical regulations. Mice were housed in a facility accredited by the Japan Act on the Welfare and Management of Animals (No. 105). All animal studies were approved by the Institutional Animal Care and Use Committee of Okayama University (Okayama, Japan), and conformed to the Japan Act on the Welfare and Management of Animals (No. 105). Littermates were used in this study to randomize genetic variation. Mice carrying a $Fktn^{flox/flox}$ allele for conditional deletion of FKTN exon 2 were generated by homologous recombination in mouse C57BL/6 ES cells[23]. To produce $Fktn^{flox/flox}$;MCK-Cre[+/−] (MCK-$Fktn$-cKO) mice, $Fktn^{flox/flox}$ mice were mated with transgenic mice expressing Cre recombinases under the control of the MCK promoter (B6.FVB(129S4)-Tg(Ckmm-cre)5Khn/J; The Jackson Laboratory, Stock No. 006475), and the offspring were back-crossed with $Fktn^{flox/flox}$ mice (C57BL/6 background) over 25 generations. Transgenic mice expressing tamoxifen-inducible cardiomyocyte-specific Cre recombinase, $\alpha$MHC-MCM[+/−] (FVB/N background; The Jackson Laboratory, Stock No. 005650), were back-crossed with C57BL/6 mice over 20 generations. To delete the floxed alleles in a tamoxifen-inducible cardiomyocyte-specific manner, $Fktn^{flox/flox}$ mice were mated with $\alpha$MHC-MCM[+/−] mice from an obtained colony, and the offspring

were back-crossed with $Fktn^{flox/flox}$ mice (C57BL/6 background) over 20 generations. As a result, we obtained C57BL/6-congenic $Fktn^{flox/flox}$;$\alpha$MHC-MCM[+/−] mice ($\alpha$MHC-MCM-$Fktn$-cKO). We used $Fktn^{flox/flox}$ littermates mainly as age-matched controls in this study. Germline transmission of $Fktn$ conditional null alleles was confirmed by Southern blotting and PCR genotyping using the primer pair, Fukutin-F, 5′-GTCAAATAGCATAATTACGGGACAG-3′ and Fukutin-R, 5′-CAAGTATGGCAGTACACATTTATCG-3′, yielding products of 778 bp (wild type) and 870 bp (null allele). Cre transgene was confirmed by PCR using primer pair, cre-newF, 5′-CCATCTGCCACCAGCCAG–3′ and cre-newR, 5′-TCGCCA TCTTCCAGCAGG-3′. To induce Cre-mediated recombination, 8 mg/kg tamoxifen or its vehicle (peanut oil) as a control was administered by i.p. injection to $Fktn^{flox/flox}$;MCM[+/−], $Fktn^{flox/+}$;MCM[+/−], and $Fktn^{flox/flox}$;MCM[−/−] mice once daily for 4 consecutive days. Colchicine or its vehicle (saline) was administered by i. p. injection from 2 d after the start of tamoxifen. The injection began with 0.4 mg/ kg and progressed to 1 mg/kg (0.4 mg/kg, 0.6 mg/kg, 0.8 mg/kg, and 1 mg/kg) to allow the mice to adjust to the drug. Then, 1 mg/kg colchicine administration continued until 10 d after the start of tamoxifen treatment. For treatments with phenylephrine and isoproterenol, micro-osmotic pumps (Azlet model 1002) were inserted subcutaneously, delivering 80 mg/kg/d phenylephrine or 20 mg/kg/d iso-proterenol for 2 weeks. Control animals were treated with saline.

**Thoracic aortic constriction surgery**. Left ventricular pressure overload was induced by thoracic aortic constriction. The 10-week-old mice (weighing 20–22 g) were anesthetized by i.p. injections of a mixture of ketamine (100 mg/kg) and xylazine (5 mg/kg). After orotracheal intubation, cannulae were connected to a volume-cycled ventilator (SN-480-7; Shimano, Japan) with room air at a tidal volume of 0.2 ml and a respiratory rate of 110 breaths per minute. The chest cavities were entered through small incisions to the second intercostal space, and the transverse aortas were constricted with 7-0 nylon strings by ligating with a blunted 27-gauge needle, which was later removed.

**Neonatal cardiomyocyte culture**. Primary cardiomyocyte cultures were prepared from ventricles of 1-d-old mice by very gentle trypsinization at room temperature using a modified method for rat neonatal hearts[46]. Briefly, hearts were rapidly removed from neonatal MCK-$Fktn$-cKO or control (floxed) mice anesthetized with an overdose of pentobarbital (30 mg/kg, i.p.). The ventricles were excised, cut into several pieces, and washed three times with 10 ml ice-cold PBS for 1 min by gentle shaking. The tissue pieces were digested three times with 0.06% trypsin in DMEM (8 ml) for 8 min at 37 °C by gentle agitation. The cells were resuspended in DMEM with 10% FCS to stop trypsinization and centrifuged at 1400 × g for 3 min. The cell pellets were resuspended in fresh DMEM containing 10% FCS, plated on collagen-coated 24-well dishes at a density of 4 × 10$^4$ cells per well, and maintained in DMEM containing 10% FCS. After 24 h, the cells were divided into three groups and then maintained for up to 72 h in DMEM alone or DMEM with 10% FCS or with 10 μM phenylephrine. The formation of myocytes clusters and spontaneous synchronized beating were confirmed by observation under an inverted microscope (CKX41; Olympus). Cellular viability was investigated at 48 h of culture by using a The LIVE/DEA Fixable Green Dead Cell Stain Kit (Invitrogen). Cells viability were validated on three independent experiments for cell type.

**Isolation of adult mouse ventricular myocytes**. Ventricular myocytes were obtained from 10-month-old male MCK-$Fknt$-cKO, $\alpha$MHC-MCM-$Fknt$-cKO, or floxed-control mice. Hearts were rapidly removed from adult mice anesthetized with an overdose of pentobarbital (300 mg/kg, i.p.) and Langendorff-perfused at a constant hydrostatic pressure of 70 cm $H_2O$ at 37 °C using cell-isolation buffer (CIB; 130 mM NaCl, 5.4 mM KCl, 0.5 mM $MgCl_2$, 0.33 mM $NaH_2PO_4$, 22 mM glucose, 50 nM/ml bovine insulin, and 25 HEPES-NaOH (pH 7.4)) supplemented with 0.4 mM EGTA, which chelates calcium within the heart. Insulin was used from a 1 U/ml stock solution in 0.1 mM HCl (pH 4.0). EGTA was from a 400 mM stock in 1 M NaOH (pH 7.8). The perfusate was then switched to the enzyme solution (15 ml) of CIB supplemented with 0.3 mM $CaCl_2$, 1 mg/ml collagenase, 0.06 mg/ml trypsin, and 0.06 mg/ml protease. Once the tissue had undergone complete digestion, the ventricles were excised, cut into several pieces, and further digested in fresh enzyme solution (15 ml) for 15–20 min at 37 °C until they were mostly dissociated. The $CaCl_2$ level in this enzyme solution was increased to 0.7 mM and supplemented with 2 mg/ml BSA. The cell suspension was centrifuged at 1400 × g for 3 min. The cell pellet (~0.1 ml) was resuspended in CIB supplemented with 1.2 mM $CaCl_2$ and 2 mg/ml BSA, then incubated at 37 °C for 10 min, centrifuged (1400 × g for 3 min), and resuspended in 10 ml Tyrode's solution (140 mM NaCl, 5.4 mM KCl, 1.8 mM $CaCl_2$, 0.5 mM $MgCl_2$, 0.33 mM $NaH_2PO_4$, 11 mM glucose, and 5 mM HEPES-NaOH (pH 7.4)) supplemented with 2 mg/ml BSA. Isolated cardiomyocytes were incubated at 37 °C in Tyrode's solution containing 10 μM colchicine for 1 h to depolymerize MTs or 25 μM ilimaquinone for 15 min to induce vesiculation of Golgi membranes.

**Cell shortening and $Ca^{2+}$ transients in isolated myocytes**. Isolated cardiomyocytes were loaded with 10 μM Indo-1 AM and electrically stimulated at 1 Hz using a two-platinum electrode insert connected to a bipolar stimulator (SEN-3301; Nihon Kohden) on the stage of an inverted microscope (IX71; Olympus) with a

20× water immersion objective lens (UApo N340; Olympus). Calcium transients were measured as the ratio of fluorescence emitted at 405/480 nm, after excitation at 340 nm, using a high-performance Evolve EMCCD camera (Photometrics). Cardiomyocytes were maintained under continuous flow in standard Tyrode's solution, exchanged using a microperfusion system. The experiments were recorded and analyzed using MetaMorph software (version 7.7.1.0; Molecular Devices). To estimate the effect of colchicine on cell contraction, we calculated the efficiency of converting elevated $Ca^{2+}$ levels to cell contraction by dividing the percentage of cell shortening by the $Ca^{2+}$ amplitude.

**SR $Ca^{2+}$ contents in adult cardiac myocytes**. SR $Ca^{2+}$ content was assessed by rapid applying 10 mM caffeine and measuring the resulting $Ca^{2+}$ transients in isolated cardiomyocytes. Fura-2 acetoxymethyl ester (Fura-2)-loaded myocytes were alternately excited at 340 and 380 nm using a Lambda DG-4 ultra-high-speed wavelength switcher (Sutter Instrument) coupled to an inverted microscope IX71 (Olympus) with a UApo 20×/0.75 objective lens (Olympus). The Fura-2 fluorescent signal was recorded by ORCA-Flash 2.8 (Hamamatsu Photonics) and analyzed using a ratiometric fluorescence method with MetaFluor software (version 7.7.5.0; Molecular Devices).

**Generation of knockout cell line with CRISPR/Cas9**. The CRISPR/Cas9 targeting sequence for *FKTN* was GAGTAGAATCAATAAGAACG. The oligonucleotides for this sequence were inserted into the Cas9 Smart nuclease all-in-one vector (System Biosciences, Mountain View, CA) with an additional 8-base sequence at the 5′ terminus. The vector was transfected into HEK293 (American Type Culture Collection, Manassas, VA, USA; CRL-1573) or HAP1 cells (Haplogen, Vienna, Austria; C631). IIH6-negative cells were sorted by fluorescence-activated cell sorting (MoFlo, Beckman Coulter, Brea, CA). HEK293 cells were cultured in DMEM supplemented with 10% FCS and penicillin/streptomycin. HAP1 cells were cultured in Iscove's modified Dulbecco's medium (Wako Pure Chemical Industries) supplemented with 10% FCS and penicillin/streptomycin. Each cell clone was verified for IIH6 reactivity by Western blot analysis and DNA sequencing. The mutations in each clone were as follows: (HEK293 cell is a triploid) HEK FKTN1o-2 (27-bp deletion, 1-bp deletion, and 1-bp insertion in exon 2, causing frameshifts) and HAP FKTN5 (34-bp deletion in exon 2, causing a frameshift).

**Histology**. Hearts were excised and immediately fixed in buffered 4% paraformaldehyde, embedded in paraffin, and sectioned at a thickness of 4 μm.

**Electron microscopy**. Excised hearts were fixed in 2% paraformaldehyde/2% glutaraldehyde in PBS. The sections were examined under a JEM-1200 electron microscope (Nihondensi, Co., Japan).

**Antibodies**. The following antibodies were used for immunostaining or immunoblot analysis: anti-IIH6 (1:500 dilution, 05–593, Millipore); anti-β-DG (1:1000 dilution, B-DG-CE, Novocastra); anti-dystrophin (1,1000 dilution, sc-15376), anti-JP2 (1:500 dilution, sc-51313), anti-calcineurin (1:1000 dilution, sc-9070), anti-CamKII (1:1000 dilution, sc-9035), and anti-phosphorylated CamKII (1:1000 dilution, sc-12886) (Santa Cruz); anti-αSG (1:500 dilution, 20A6), anti-βSG (1:500 dilution, 5B1), and anti-γSG (1:500 dilution, 2185) (Leica); anti-Akt (1:1000 dilution, 9272), anti-phosphorylated Akt (1:1000 dilution, 4060), anti-PKD (1:1000 dilution, 2052), anti-phosphorylated PKD (S916, S744) (1:1000 dilution, 2051 and 2054, respectively), anti-HDAC9 (1:1000 dilution, 7628), and anti-MEF2 (1:1000 dilution, 5030) (Cell Signaling); anti-GAPDH (1:2000 dilution, ab9484), anti-MCIP (1:1000 dilution, ab25124), and anti-GM130 (1:500 dilution, ab52649) (Abcam); anti-phosphorylated HDAC9 (1:1000 dilution, SAB4300269) and anti-α-tubulin (1:100 dilution, T5168) (Sigma); anti-PI4P (1:500 dilution, Z-P004) (Echelon); anti-Na$^+$/K$^+$ ATPase (1:1000 dilution, 07–674), anti-SERCA (1:1000 dilution, MA3-910), and anti-RyR (1:1000 dilution, MA3-916) (Thermo); anti-ANP (1;500 dilution, H005-24, Phoenix); anti-Cav3 (1:1000 dilution, 610420, BD Pharmingen); and anti-LTCC (1:1000 dilution, AC-003, Alomone). The anti-NCX1 antibody was generated in our laboratory. Anti-core α-DG is previously reported[64].

**Immunocytochemistry**. Frozen heart sections (5 μm), embedded in OCT compound (Tissue-Tek), were permeabilized with 0.1% Triton X-100 and incubated with primary antibodies. Mouse On Mouse (M.O.M.) Blocking Reagent (Vector Laboratories) was used to block endogenous mouse antibody in the tissue sections. Cells were examined using a confocal microscope (Fluoview FV1000; Olympus) mounted on an Olympus IX81 epifluorescence microscope with a UPlanSApo 60×/ 1.35 oil immersion objective lens (Olympus).

**Immunoblotting**. Mice hearts were homogenized using a Hiscotron homogenizer (NITI-ON) in lysis buffer containing 20 mM HEPES (pH 7.4), 150 mM NaCl, 1% sodium deoxycholate, 1% SDS, 2 μg/ml leupeptin, 1 μg/ml aprotinin, 200 μM phenylmethylsulfonyl fluoride, and 200 μM benzamidine hydrochloride. The lysates were centrifuged at $100,000 \times g$ for 20 min and the supernatants were used for immunoblot

analysis. DG from solubilized myocardium was enriched with wheat germ agglutinin-agarose beads (Vector Laboratories). To detect the expression of FKTN protein, cardiac muscles were solubilized with 1% Triton X-100 in TBS. The solubilized samples were assayed for total protein concentration and then 4 mg of proteins were subjected to FKTN antibody-conjugated beads to enrich FKTN proteins as previously reported[65]. The bound materials were eluted with 0.1 M glycine-HCl (pH 2.5) and then analyzed by Western blotting using FKTN antibody (RY213). Immunoreactive bands were visualized using a chemiluminescence detection system (Perkin Elmer or Amersham Biosciences Corp.) and an LAS3000 luminescent image analyzer (Fuji Film). Uncropped and unprocessed full scan images are shown in Supplementary information (Supplementary Fig. 14).

**Microarray analysis**. Microarray analysis was performed with an Agilent gene expression microarray platform (SurePrint G3 Mouse Gene Expression 8 × 60 K). We visualized genes with the largest variance using a hierarchical clustering heatmap. For each gene, we compared expression levels between floxed and *Fktn*-cKO RNAs. Gene expression differences were evaluated using the eBayes function in the limma package[66], microarray data were fit to a linear model, and differential expression was estimated by empirical Bayes moderation. The resulting $P$-values were corrected via the Benjamini and Hochberg method using the topTable function. Differentially expressed genes were defined as those with $Log_2$ changes of at least 1.5-fold between a pair of samples at an FDR of 0.001 for genes. Supplementary Table 1 provides the complete dataset. For differentially expressed genes, we carried out functional annotation analysis using DAVID. Differentially expressed genes were used as the input gene list, and all mouse genes that were expressed in the heart were used as the background. We looked for enrichment of genetic associations within KEGG pathways. Enrichment pathways were analyzed using ReactomePA.

**Data analysis**. Data were analyzed by individuals who were blinded to the genotype, drug treatment, and operation. Data presented here were reproducible in at least three independent experiments. Results are shown as the means ± s.e. m. Paired data were evaluated using a Student's $t$-test. For multiple comparisons, analyses of variance with Tukey–Kramer tests were performed as appropriate using GraphPad Prism. The Kaplan–Meier method, with a log-rank test, was used for survival analysis. A $P$ value of $< 0.05$ was considered statistically significant.

**Reporting summary**. Further information on research design is available in the Nature Research Reporting Summary linked to this article.

## Data availability

The data that support the findings of this study are available from the corresponding author upon reasonable request. The source data underlying Figs. 1e, 3c, 4c, 4e, and 8b, d are provided as Source Data file. The gene expression data were deposited at NCBI's Gene Expression Omnibus (GEO). It is accessible through GEO series accession number of GSE138280.

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

## Acknowledgements

This work was supported by Grants-in-Aid from the Ministry of Education, Culture, Sports, Science, and Technology of Japan (grant numbers 17H02085, 18K19924 to Y.K. and 17H04740 to Y.U.), the Japan Society for the Promotion of Science Funding Program for Next Generation World-Leading Researchers (grant number 10104401 to Y.K.), the Japan Agency for Medical Research and Development (grant numbers JP17gm5810003 to Y.K. and JP17gm0810010 to M.K.), the National Center of Neurology and Psychiatry (Intramural Research Grant 29-4 to T.T.) and Grant-in-Aid for scientific Research on Innovative Areas (JP17H06421 to M.K.), MEXT. This work was supported in part by a

grant from the Mochida Memorial Foundation for Medical and Pharmaceutical Research, SENSHIN Medical Research Foundation, Suzuken Memorial Foundation, Wesco scientific promotion foundation, and a Shiseido Female Researcher Science Grant to Y.K. and Takeda Science Foundation to M.K. and Y.K.

## Author contributions

Y.U. and Y.K. planned and designed the study, performed the majority of experiments, and wrote the manuscript. M.K. and Y.K. generated the FKTN-deficient mice. M.K. contributed to detection of DGC and α-DG glycosylation and discussed the data. S.M. takes responsibility for cardiac physiology. S.T. carried out the immunofluorescence experience. K.K. generated FKTN-deficient HEK293 and HAP cell lines. K.N. and T.T. provided conceptual advice.

## Competing interests

The authors declare no competing interests.
