## [Peer Review File · Nature Communications]

Reviewers' Comments:

Reviewer #1:

Remarks to the Author:

Fukutin is a gene whose mutation causes Fukuyama-type congenital muscular dystrophy (FCMD). Heart failure is a major cause of death in patients with this type of muscular dystrophy. This manuscript by Ujihara and colleagues investigated the cellular and molecular mechanisms of heart failure associated with muscular dystrophy using cardiac-specific fukutin gene knockout mouse models, including an MCK promoter-driven conventional knockout and an α MHC-MerCreMer-mediated cardiac inducible fukutin knockout. These two models present strikingly different phenotypes: cardiac dysfunction was not present until 6 months or older in the chronic fukutin knockout model, while acute fukutin knockout resulted into severe cardiac dysfunction and sudden cardiac death within a few days after tamoxifen induction. At the cellular and molecular levels, the chronic model is associated with normal cardiac structure and function in early adulthood (i.e. at or before 16 months). Although fukutin deficiency led to a severe disruption in the dystrophin-glycoprotein complex early (6 weeks of age), this finding raised the question of whether and how important the dystrophin-glycoprotein complex is to cardiomyocyte cell function at baseline. Interestingly, from the acute model, the authors discovered that the microtubule / junctophilin-2 / T-tubule pathway is critically involved in the pathophysiology of heart failure induced by acute fukutin deficiency, which is independent from the dystrophin-glycoprotein complex. In fukutin-deficient cardiomyocytes isolated from the acute depletion model, abnormal accumulation of microtubules, junctophilin-2 downregulation and T-tubule structural disorganization were observed. Administration of colchicine, a microtubule depolymerizer, was able to correct these defects in cardiomyocyte subcellular structural organization. The authors also presented in vivo data that colchicine treatment prevented fukutin deficiency-induced heart dysfunction. Additional data reported in this paper demonstrate that fukutin-deficient hearts are vulnerable to hypertrophic stress such as TAC, isoproterenol or phenylephrine.

This is an important study which revealed for the first time a likely mechanistic connection between Golgi apparatus and myocyte T-tubule organization through microtubules. One translational insight drawn from this study is that intervention of microtubule dynamics may be promising for treating α -DGpathy-associated cardiomyopathies.

Fukutin is a Golgi-localized protein and is involved in the pathway of protein glycosylation. How does fukutin regulate Golgi structure and function and thus lead to microtubule accumulation? This seems to be an important and open question remaining in this manuscript. Further evidence that provides more insight into this question will improve the manuscript. However, this reviewer recognizes that this is a challenging issue to be addressed.

For the acute colchicine experiments (Figure 6e/f), colchicine treatment for 1 hour is shown to improve contractile function in $fktn^{-/-}$ cardiomyocytes. What are the underlying changes responsible for this fast recovery of contractile function?

Previous work by another group (PMC4006305) suggests that microtubule accumulation regulates the distribution of junctophilin-2 protein instead of its expression level. The authors only provided evidence by Western blot (Figure 6a) that junctophilin-2 is downregulated in acute fukutin-deficient hearts. Do the authors think that microtubule densification contributes to junctophilin-2 downregulation? Is junctophilin-2 distribution altered in this model?

Reviewer #2:

Remarks to the Author:

The manuscript aims to understand the molecular patho-mechanism of cardiac failure in muscular dystrophy, specifically in dystroglycanopathy using cardiac-specific $Fktn$ knockout

mice as a model system. This is critically important for improving life quality and life span of all MD patients as the authors stated in the manuscript. The authors reported their main findings of the study including the demonstration that FKTN was crucial for maintaining contractile function and Ca²⁺ handling. Another important finding of the study is that FKTN is essential for maintaining the Golgi-MT network in cardiomyocytes. The authors consider that the discovered mechanisms could help to develop new therapies to interfere the pathways implicated in the disease progression.

While the observation of the study is valid, the reviewer has 2 critical questions about the animal models.

1). In the Results, the authors describe the expression of functionally glycosylated a-DG with IIH6 in the flox/flox control mice undetectable until 24 weeks. This is contradictory to the referenced publication (23) describing the detection of glycosylated a-DG in the same type of floxed control mice as normal. Further, the Supplementary Fig 1 showed negative a-DG in the vehicle animal. However, the expression of FKTN shown in the Supplementary Fig 3 is normal at 10 weeks of age. What is the explanation for the negative functionally glycosylated a-DG, but a normal FKTN expression?

2). The treatment of 10-week-old MHC-MCM-Fktn-cKO mice for 4 days with tamoxifen induced chamber dilation and severe cardiac dysfunction with ~50% death rate within 1 week. So the question is if the parental mice do not express functionally glycosylated a-DG at 10 weeks at the first place, then the whole event is most likely unrelated to the FKTN expression and the lack functionally glycosylated a-DG. Further, if this is related to the lack of b-DG only (shown in Sup Fig 4) triggered by the tamoxifen treatment, then the whole event could be unrelevant to what occur in the diseased muscle with known Fktn mutation which does not specifically lack of b-DG expression.

The authors need to discuss these issues carefully.

Reviewer #3:

Remarks to the Author:

The authors present an elegant study of the role of FKTN in the heart. They show in cardiac-specific Fktn KO mice that FKTN has a crucial role in the maintenance of myocyte structure and contractile function. The study offers significant novel insight into the potential molecular mechanisms involved in FKTN-deficient cardiomyopathy, and tantalisingly, suggests a potential new therapy, using a drug which already has FDA approval. The study will be of significant interest to the neuromuscular disorder field, and more widely to the cardiomyopathy field. The authors use standard statistical methods which appear to be appropriate to the study.

Although the study is, on the whole, very well executed, I have three criticisms which I would invite the authors to respond to:

1. I would be critical of the data from the cross sectional area of cardiomyocytes (Fig 4a). How can we be sure that the control cells are not adhering more efficiently to the plate and therefore becoming more flattened and wider than the KO cells? 3D reconstruction of cells and measurement of total volume, although time consuming, may be a more reliable method of looking at hypertrophy.
2. For the same experiment, the authors have not indicated the level of cell viability before and after treatment with PE- if the KO cells are not responding by undergoing hypertrophy, are more dying?
3. The observation that the KO cells do not undergo spontaneous hypertrophy (at 48h in the absence of PE) like the control cells is an interesting one- the authors have not made comment on this.
4. The authors have not made clear the purity of their cardiomyocyte populations (I would expect there to be some fibroblast component). Was the purity comparable in the control and KO mice?

This has impact on the interpretation of protein data.

5. It is difficult to conclude if the observation of changes in golgi-MT morphology are specific to Fukitin deficiency without the presence of disease controls for these experiments. The authors have however eluded to this in the final sentence of their discussion.

Minor comment

6. Figure 3- panels b,c,d and e don't appear to be in a logical order

Responses to reviewers' comments

We would like to thank the reviewers for their careful consideration of our manuscript. We believe that their suggestions have greatly improved our manuscript. Please find the point-by-point responses to their comments below. The changes in the manuscript are in red font.

Reviewer #1 (Remarks to the Author):

Fukutin is a gene whose mutation causes Fukuyama-type congenital muscular dystrophy (FCMD). Heart failure is a major cause of death in patients with this type of muscular dystrophy. This manuscript by Ujihara and colleagues investigated the cellular and molecular mechanisms of heart failure associated with muscular dystrophy using cardiac-specific fukutin gene knockout mouse models, including an MCK promoter-driven conventional knockout and an α MHC-MerCreMer-mediated cardiac inducible fukutin knockout. These two models present strikingly different phenotypes: cardiac dysfunction was not present until 6 months or older in the chronic fukutin knockout model, while acute fukutin knockout resulted into severe cardiac dysfunction and sudden cardiac death within a few days after tamoxifen induction. At the cellular and molecular levels, the chronic model is associated with normal cardiac structure and function in early adulthood (i.e. at or before 16 months). Although fukutin deficiency led to a severe disruption in the dystrophin–glycoprotein complex early (6 weeks of age), this finding raised the question of whether and how important the dystrophin–glycoprotein complex is to cardiomyocyte cell function at baseline. Interestingly, from the acute model, the authors discovered that the microtubule / junctophilin-2 / T-tubule pathway is critically involved in the pathophysiology of heart failure induced by acute fukutin deficiency, which is independent from the dystrophin–glycoprotein complex. In fukutin-deficient cardiomyocytes isolated from the acute depletion model, abnormal accumulation of microtubules, junctophilin-2 downregulation and T-tubule structural disorganization were observed. Administration of colchicine, a microtubule depolymerizer, was able to correct these defects in cardiomyocyte subcellular structural organization. The authors also presented in vivo data that colchicine treatment prevented fukutin deficiency-induced heart dysfunction. Additional data reported in this paper demonstrate that fukutin-deficient hearts are vulnerable to hypertrophic stress such as TAC, isoproterenol or phenylephrine.

This is an important study which revealed for the first time a likely mechanistic connection between Golgi apparatus and myocyte T-tubule organization through microtubules. One translational insight drawn from this study is that intervention of microtubule dynamics may be promising for treating α -DGpathy-associated cardiomyopathies.

Fukutin is a Golgi-localized protein and is involved in the pathway of protein glycosylation. How does fukutin regulate Golgi structure and function and thus lead to microtubule accumulation? This seems to be an important and open question remaining in this manuscript. Further evidence that provides more insight into this question will improve the manuscript. However, this reviewer recognizes that this is a challenging issue to be addressed.

Response: We agree with the reviewer's suggestion and have added new data, now presented in Fig. 8c and described under "*MT depolymerization improves contractility in Fktn-deficient cardiomyocytes*" (p. 16) and in the Discussion (p. 19–20). We found that PKD phosphorylation levels were significantly higher in α MHC-MCM-*Fktn*-cKO hearts treated with tamoxifen for 4 days, indicating a role of FKTN in cardiomyocytes in PKD activity, which is known to regulate Golgi-to-cell surface transport by controlling the biogenesis of specific transport carriers and Golgi fragmentation. Thus, we think that the activation of PKD by acute *Fktn* elimination, along with an impairment of acute glycosylation at the Golgi apparatus, contributed to Golgi fragmentation in *Fktn*-deficient hearts. We also suggest that the altered Golgi morphology was affected by MT densification.

For the acute colchicine experiments (Figure 6e/f), colchicine treatment for 1 hour is shown to improve contractile function in fktn-/- cardiomyocytes. What are the underlying changes responsible for this fast recovery of contractile function?

Response: To examine the mechanisms by which the acute colchicine treatment improved contractility in *Fktn*-deficient myocytes, we measured the intracellular Ca^{2+} handling during E-C coupling and analyzed contractile efficiency (shortening per Ca^{2+} amplitude), which are now presented in Fig. 8d and described in the Results (page 16) and the Discussion (p. 20–21). We found that colchicine increased the amplitude of intracellular Ca^{2+} transients during E-C coupling and improved the contractile efficiency. Thus, the depolymerization of MTs in *Fktn*-deficient myocytes enhanced systolic function by increasing Ca^{2+} transients and decreasing the mechanical resistance to shortening.

Previous work by another group (PMC4006305) suggests that microtubule accumulation regulates the distribution of junctophilin-2 protein instead of its expression level. The authors only provided evidence by Western blot (Figure 6a) that junctophilin-2 is downregulated in acute fukutin-deficient hearts. Do the authors think that microtubule densification contributes to junctophilin-2 downregulation? Is junctophilin-2 distribution altered in this model?

Response: We thank the reviewer for this insightful comment. We have added new results (p. 14),

presented in Fig. 7g, showing that the JP2 distribution is altered in α MHC-MCM-*Fktn*-cKO hearts. As previously reported by Zhang et al (Circulation, 2014), there is a possibility that the MT-mediated defects in JP2 trafficking contribute to T-tubule remodeling in *Fktn*-deficient myocytes. We address this in the final paragraph of the Discussion in the revised manuscript.

Reviewer #2 (Remarks to the Author):

The manuscript aims to understand the molecular patho-mechanism of cardiac failure in muscular dystrophy, specifically in dystroglycanopathy using cardiac-specific Fktn knockout mice as a model system. This is critically important for improving life quality and life span of all MD patients as the authors stated in the manuscript. The authors reported their main findings of the study including the demonstration that FKTN was crucial for maintaining contractile function and Ca²⁺ handling. Another important finding of the study is that FKTN is essential for maintaining the Golgi-MT network in cardiomyocytes. The authors consider that the discovered mechanisms could help to develop new therapies to interfere the pathways implicated in the disease progression.

While the observation of the study is valid, the reviewer has 2 critical questions about the animal models.

1). In the Results, the authors describe the expression of functionally glycosylated α -DG with IIIH6 in the flox/flox control mice undetectable until 24 weeks. This is contradictory to the referenced publication (23) describing the detection of glycosylated α -DG in the same type of floxed control mice as normal. Further, the Supplementary Fig 1 showed negative α -DG in the vehicle animal. However, the expression of FKTN shown in the Supplementary Fig 3 is normal at 10 weeks of age. What is the explanation for the negative functionally glycosylated α -DG, but a normal FKTN expression?

Response: In this study, we generated two lines of *Fktn* conditional knockout mice, namely, the MCK promoter-driven cKO and the temporally controlled and cardiac-specific α MHC-MCM-*Fktn*-cKO. As the reviewer points out, we can detect glycosylated α -DG at all developmental stages in the skeletal muscle of the MCK-promoter driven cKO mice by immunoblotting (Kanagawa et al., Hum Mol Genet, 2013). In this study, we used a different method (immunofluorescence) to analyze the levels of glycosylated α -DG in heart tissue (Fig. 1a). The roles of glycosylated α -DG may differ between in cardiomyocytes and skeletal muscle. Moreover, we detected weak immunoreactivity in cardiac tissue at 6–16 weeks, which may be detectable by immunoblotting with a concentrated protein extract. It is important to note that the immunoreactivities of glycosylated α -DG and DGC increases in the sarcolemma at 24 weeks after birth in normal control mice.

Supplementary Fig. 1 shows immunoblots for FKTN and glycosylated α -DG in MCK promoter-driven cKO and control mice, whereas Supplementary Fig. 3 shows an immunoblot for FKTN in the cardiac-specific α MHC-MCM-*Fktn*-cKO with or without tamoxifen. As functional FKTN is deleted after birth in cardiomyocytes of MCK-promoter driven cKO mice, the protein is not detected at a young age (Supplementary Fig. 1). In α MHC-MCM-*Fktn*-cKO mice, FKTN is detected in the absence of tamoxifen and decreases with treatment (Supplementary Fig. 3).

2). The treatment of 10-week-old MHC-MCM-Fktn-cKO mice for 4 days with tamoxifen induced chamber dilation and severe cardiac dysfunction with ~50% death rate within 1 week. So the question is if the parental mice do not express functionally glycosylated a-DG at 10 weeks at the first place, then the whole event is most likely unrelated to the FKTN expression and the lack functionally glycosylated a-DG. Further, if this is related to the lack of b-DG only (shown in Sup Fig 4) triggered by the tamoxifen treatment, then the whole event could be unrellevant to what occur in the diseased muscle with known Fktn mutation which does not specifically lack of b-DG expression. The authors need to discuss these issues carefully.

Response: In our study, *Fktn* was eliminated in 10-week-old MHC-MCM-*Fktn*-cKO mice by treating them with tamoxifen for 4 days. Animals not treated with tamoxifen (including parental mice) show normal FKTN expression.

We have modified the figures in the revised manuscript. As this reviewer suggested, we show that the treatment of 10-week-old MHC-MCM-*Fktn*-cKO mice for 4 days with tamoxifen induced chamber dilation and severe cardiac dysfunction with ~50% death rate within 1 week (revised Fig. 6). Figures 6–8 now show the results of our analyses of the structure and function of myocytes in these mice. Supplementary Fig. 4 (middle panels) shows that there is normal expression and localization of glycosylated α -DG and DGC proteins at sarcolemma in cKO hearts after 4 days of tamoxifen treatment, which decline after 8 days after tamoxifen. These observations suggest that cardiac dysfunction at 4 days after tamoxifen injection is independent of α -DG glycosylation and DGC protein in sarcolemma (see page 12 of the revised manuscript).

Reviewer #3 (Remarks to the Author):

The authors present an elegant study of the role of FKTN in the heart. They show in cardiac-specific Fktn KO mice that FKTN has a crucial role in the maintenance of myocyte structure and contractile function. The study offers significant novel insight into the potential molecular mechanisms involved in FKTN-deficient cardiomyopathy, and tantalisingly, suggests a potential new therapy, using a drug which already has FDA approval. The study will be of significant interest to the neuromuscular

disorder field, and more widely to the cardiomyopathy field. The authors use standard statistical methods which appear to be appropriate to the study.

Although the study is, on the whole, very well executed, I have three criticisms which I would invite the authors to respond to:

1. I would be critical of the data from the cross sectional area of cardiomyocytes (Fig 4a). How can we be sure that the control cells are not adhering more efficiently to the plate and therefore becoming more flattened and wider than the KO cells? 3D reconstruction of cells and measurement of total volume, although time consuming, may be a more reliable method of looking at hypertrophy.

Response: We agree with the reviewer and understand that cellular adhesiveness affects the cross-sectional area. Neonatal cardiomyocytes from cKO mice show a defect in cellular extensibility rather than an impairment of cell adhesion. In the revised manuscript, we have included new results (p. 9), which are depicted in Fig. 4. In particular, Fig. 4b shows the magnified view of phalloidin staining in cKO cells, revealing immature and tangled myofilaments, suggesting that the defect is in cellular extensibility.

2. For the same experiment, the authors have not indicated the level of cell viability before and after treatment with PE- if the KO cells are not responding by undergoing hypertrophy, are more dying?

Response: To address the reviewer's suggestion, we included new results (p. 9 and Fig. 4d) showing that living cardiomyocytes treated with PE show caffeine-induced Ca^{2+} release from the SR (a typical hypertrophic response). The release of Ca^{2+} in the presence of PE was observed in cKO myocytes, indicating that they are viable but only lack a functional hypertrophic response.

3. The observation that the KO cells do not undergo spontaneous hypertrophy (at 48h in the absence of PE) like the control cells is an interesting one- the authors have not made comment on this.

Response: To examine why the hypertrophic response was lost in cKO cells, we analyzed the localization of MEF2 and HDAC9 with or without PE-treatment (Fig. 5a and b), which are transcriptional regulators in hypertrophic cardiomyocyte (see page 10). These observations suggest that MEF2 and HDAC9 translocation are impaired in *Fktn*-deficient myocytes, resulting in the loss of hypertrophic responses.

4. The authors have not made clear the purity of their cardiomyocyte populations (I would expect there to be some fibroblast component). Was the purity comparable in the control and KO mice? This has impact on the interpretation of protein data.

Response: We can easily distinguish cardiomyocytes from fibroblasts in our cultures by their morphology. Staining with phalloidin revealed that > 90 % of the cells were cardiomyocytes. We added new results pertaining to fibroblasts, which are shown in Fig. 4d. Specifically, cardiomyocytes, but not fibroblasts, show the caffeine-induced Ca^{2+} release. We did not include protein data from these cells.

Figure 3f shows the change in protein expression induced by a hypertrophic intervention in MCK promoter-driven cKO hearts. We used GAPDH for the loading control. The effect on fibrosis is shown in Fig. 3d.

5. It is difficult to conclude if the observation of changes in golgi-MT morphology are specific to Fukitin deficiency without the presence of disease controls for these experiments. The authors have however eluded to this in the final sentence of their discussion.

Response: We agree with this reviewer's suggestion. In the Discussion, we mention that studies show MT-mediated defects in T-tubule organization in failing human hearts and in an animal model of heart failure (p. 20–21). For example, skeletal muscles in *mdx* mice (dystrophic model) show abnormal Golgi morphology and distribution (Percival and Froehner, Traffic, 2007). In addition, we analyzed the phosphorylation of PKD, which affects Golgi morphology, in the hearts of *mdx* mice of several ages (see figure below). We found that PKD phosphorylation increases with age in *mdx* hearts compared with that in control mice (C57BL6). However, we cannot conclude how this increase relates to the onset of cardiomyopathy in *mdx* mice. To elucidate whether changes in Golgi-MT morphology are common in other diseases, we need to analyze the onset of disease in other models at all developmental stages or generate controlled transient cKO models. Nevertheless, the depolymerization of MTs by colchicine is a common strategy for therapeutic intervention in muscular dystrophy-associated heart failure. Moreover, a previous study reported that colchicine treatment improved Ca^{2+} handling in the hearts of old *mdx* mice (Kurt et al., JACC Basic Transl Sci, 2016). Further investigations are needed to clarify how MTs link the Golgi apparatus and myocyte T-tubule organization. Our data suggest that FKTN maintains MT integrity; whether this is preserved in other animal models also needs further investigation.

Phosphorylation level of PKD in c57BL/6 mice or mdx mice ($n = 4$). Data are means \pm s.e.m. * $P < 0.05$ versus pretreatment; # $P < 0.05$ between indicated groups based on Tukey-Kramer tests.

Minor comment

6. Figure 3- panels b,c,d and e don't appear to be in a logical order

Response: We have modified the text in the revised manuscript (p. 7).

Reviewers' Comments:

Reviewer #1:

Remarks to the Author:

The authors have satisfactorily addressed all my concerns in this revision.

Reviewer #2:

Remarks to the Author:

Original comment 1). In the Results, the authors describe the expression of functionally glycosylated α -DG with IIH6 in the flox/flox control mice undetectable until 24 weeks. This is contradictory to the referenced publication (23) describing the detection of glycosylated α -DG in the same type of floxed control mice as normal. Further, the Supplementary Fig 1 showed negative α -DG in the vehicle animal.

However, the expression of FKTN shown in the Supplementary Fig 3 is normal at 10 weeks of age. What is the explanation for the negative functionally glycosylated α -DG, but a normal FKTN expression?

Response: As the reviewer points out, we can detect glycosylated α -DG in skeletal muscle of MCK promoter-driven cKO mice at all developmental stages by immunoblotting (Kanagawa et al., HumMol Genet, 2013). However, in our manuscript, we showed the expression level of glycosylated α -DG in heart tissue (i.e., immunofluorescence; Fig. 1a). These observations suggest that the roles of glycosylated α -DG differ between the heart and skeletal muscle. (In addition, the levels of glycosylated α -DG are detectable in 10-weeks hearts by immunoblotting (supplementary Fig. 1b). Therefore, we can detect the difference of the amount of glycosylation between floxed and cKO hearts.)

In the study described in our manuscript, we generated two lines of Fktn cKO mice: MCK promoter-driven cKO mice and temporally controlled and cardiac-specific (MHC-MCM-Fktn-cKO mice. Supplementary Fig. 1 shows immunoblots for FKTN and glycosylated α -DG in MCK promoter-driven cKO and control mice, whereas Supplementary Fig. 3 shows an immunoblot for FKTN in cardiac-specific (MHC-MCM-Fktn-cKO mice with or without tamoxifen treatment. As functional FKTN is deleted after birth in cardiomyocytes of MCK promoter-driven cKO mice, the protein is not detected at a young age (Supplementary Fig. 1). In (MHC-MCM-Fktn-cKO mice, FKTN is detected in the absence of tamoxifen and decreases with treatment (Supplementary Fig. 3

Further comment:

Indeed, the authors showed, in both the Figure 1 and SI Figure 1, that there is hardly any expression of glycosylated form of α -DG in the MCK promoter-driven cKO mice. There is no expression of FKTN either. But the isolated cardiomyocytes from these mice showed normal contractility and Ca²⁺ handling during E-C coupling. Most relevant to the claim from this study is the T-tubule disorganization described in the Tamoxifen treated FKTN-deficient cardiomyocytes. If this is to be relevant to clinic, then this feature should also be demonstrated in at least certain degree in the MCK promoter-driven cKO mice.

Original comment 2). The treatment of 10-week-old MHC-MCM-Fktn-cKO mice for 4 days with tamoxifen induced chamber dilation and severe cardiac dysfunction with ~50% death rate within 1 week. So the question is if the parental mice do not express functionally glycosylated α -DG at 10 weeks at the first place, then the whole event is most likely unrelated to the FKTN expression and the lack functionally glycosylated α -DG. Further, if this is related to the lack of β -DG only (shown in Sup Fig 4) triggered by the tamoxifen treatment, then the whole event could be unrelevant to what occur in the diseased muscle with known Fktn mutation which does not specifically lack of β -DG expression. The authors need to discuss these issues carefully.

Response: In our study, Fktn was eliminated in 10-week-old MHC-MCM-Fktn-cKO mice by treatment with tamoxifen for 4 days. Animals not treated with tamoxifen (including parental mice)

showed normal FKTN expression. As suggested by the reviewer, we show that treatment of 10-week-old MHC-MCM-Fktn-cKO mice for 4 days with tamoxifen induced chamber dilation and severe cardiac dysfunction with a ~50% death rate within 1 week (Fig. 6). Severe cardiac dysfunction in tamoxifen-treated cKO mice is triggered by a decline in FKTN expression (Supplementary Fig. 3c). In the revised manuscript, Fig. 6–8 show the results of our analyses of myocyte structure and function in these mice. Supplementary Fig. 4 (middle panels) shows that expression and localization of glycosylated α -DG and DGC proteins including β -DG at sarcolemma in cKO hearts is normal after 4 days of tamoxifen treatment and declines by 8 days after treatment. These observations suggest that the cardiac dysfunction observed after 4 days of tamoxifen treatment (Fig. 6b) is independent of α -DG glycosylation and DGC proteins in sarcolemma (p. 12). Thus, we suggest that lack of FKTN protein per se affects myocyte structure and function in cKO hearts after 4 days of tamoxifen treatment and is the trigger of pathological remodeling of cKO hearts observed 8 days after treatment.

Further comment:

The authors responded to the question 2 by saying that “Fktn was eliminated in 10-week-old MHC-MCM-Fktn-cKO mice by treatment with tamoxifen for 4 days”. Where is the evidence that tamoxifen treatment can be so swift and effective to eliminate the FKTN protein within 4 days or even 7 days? What is the half life of FKTN mRNA and protein? the only data presented in the SI Fig 3c is the reduction of FKTN, probably at 50% of normal level by WB. Further, there is no description about how many days the samples were collected after Tamoxifen treatment (although described with 4 day Tam treatment) in the Figure legend. It is hard to image that all the described changes in MT and the death of mice can be attribute to 50% reduction of FKTN. Therefore, it is critical to demonstrate the changes in levels of FKTN within a week after Tamoxifen treatment if the authors want to claim that the observed phenomenon is the result of “elimination of the FKTN”. Without such evidence, all the conclusions can hardly be sustained.

Reviewer #4:

Remarks to the Author:

The authors carried out detailed molecular and cellular analyses using two cardiac-specific, conditional Fktn knockout mouse models. They reported that microtubule densification may underlie the pathogenesis of cardiac dysfunction in the absence of Fktn and that treatment with microtubule depolymerization drug can improve the contractility in Fktn-deficient cardiomyocytes. I did not review the first version of the manuscript and was asked to comment on the response to Reviewer 3's comments. I think the authors have sufficiently addressed the questions raised by Reviewer 3.

Responses to reviewers' comments

Reviewers' comments:

Reviewer #1 (Remarks to the Author):

The authors have satisfactorily addressed all my concerns in this revision.

Response: We thank the reviewer for his thoughtful and thorough consideration of our manuscript, and for his/her positive comments.

Reviewer #2 (Remarks to the Author):

Original comment 1). In the Results, the authors describe the expression of functionally glycosylated α -DG with I1H6 in the flox/flox control mice undetectable until 24 weeks. This is contradictory to the referenced publication (23) describing the detection of glycosylated α -DG in the same type of floxed control mice as normal. Further, the Supplementary Fig 1 showed negative α -DG in the vehicle animal. However, the expression of FKTN shown in the Supplementary Fig 3 is normal at 10 weeks of age. What is the explanation for the negative functionally glycosylated α -DG, but a normal FKTN expression?

Response: As the reviewer correctly pointed out, we can detect glycosylated α -DG in the skeletal muscle of MCK promoter-driven cKO mice at all developmental stages by immunoblotting (Kanagawa et al., HumMol Genet, 2013). However, in our manuscript, we showed the expression levels of glycosylated α -DG in heart tissue (i.e., immunofluorescence; Fig. 1a). These observations suggest that the roles of glycosylated α -DG differ between the heart and skeletal muscle. (In addition, the levels of glycosylated α -DG are detectable in 10-weeks hearts by immunoblotting (supplementary Fig. 1b). Therefore, we can detect the difference in the amount of glycosylation between floxed and cKO hearts.)

In the study described in our manuscript, we generated two lines of Fktn cKO mice: MCK promoter-driven cKO mice and temporally controlled, cardiac-specific α MHC-MCM-Fktn-cKO mice. Supplementary Fig. 1 shows immunoblots for FKTN and glycosylated α -DG in MCK promoter-driven cKO and control mice, whereas Supplementary Fig. 3 shows an immunoblot for FKTN in cardiac-specific α MHC-MCM-Fktn-cKO mice with or without tamoxifen treatment. As functional FKTN is deleted after birth in cardiomyocytes of MCK promoter-driven cKO mice, the protein is not detected at a young age (Supplementary Fig. 1). In α MHC-MCM-Fktn-cKO mice, FKTN is detected in the absence of tamoxifen and decreases with treatment (Supplementary Fig. 3)

Further comment:

Indeed, the authors showed, in both the Figure 1 and SI Figure 1, that there is hardly any expression of glycosylated form of α -DG in the MCK promoter-driven cKO mice. There is no expression of FKTN either. But the isolated cardiomyocytes from these mice showed normal contractility and Ca²⁺ handling during E-C coupling. Most relevant to the claim from this study is the T-tubule disorganization described in the Tamoxifen treated FKTN-deficient cardiomyocytes. If this is to be relevant to clinic, then this feature should also be demonstrated in at least certain degree in the MCK promoter-driven cKO mice.

Response: In revised manuscript, we have added a new figure (Figure S8a), showing the abnormal distribution of Golgi and MT-accumulation in MCK promoter-driven cKO mice at old age (10 months after birth). In addition, slight T-tubule disorganization was observed in MCK-cKO hearts at this stage (Figure S8b). Our observations suggest that FKTN-deficiency leads to the disorganization of Golgi-MT networks and T-tubule structure in later adulthood in MCK promoter-driven cKO mice. We have added this discussion in the revised manuscript (p. 26).

In our study, cardiac dysfunction of MCK promoter-driven cKO mice was only observed in late adulthood. In addition, MCK promoter-driven cKO mice showed normal physiology, but were vulnerable to pathological hypertrophic stress at a young age. These observations suggest that FKTN is critical for the maintenance of cardiac structure and function under condition of stress (triggered by aging and hemodynamic stress). Therefore, MCK promoter-driven cKO hearts may have an unknown compensatory mechanism that prevents mice from manifesting any obvious phenotype of FKTN-deficiency during young adult age under normal physiological condition. We have added this discussion in the revised manuscript (p. 22-23).

Original comment 2). The treatment of 10-week-old MHC-MCM-Fktn-cKO mice for 4 days with tamoxifen induced chamber dilation and severe cardiac dysfunction with ~50% death rate within 1 week. So the question is if the parental mice do not express functionally glycosylated a-DG at 10 weeks at the first place, then the whole event is most likely unrelated to the FKTN expression and the lack functionally glycosylated a-DG. Further, if this is related to the lack of b-DG only (shown in Sup Fig 4) triggered by the tamoxifen treatment, then the whole event could be unrelevant to what occur in the diseased muscle with known Fktn mutation which does not specifically lack of b-DG expression. The authors need to discuss these issues carefully.

Response: In our study, FKTN expression was eliminated in 10-week-old α MHC-MCM-Fktn-cKO mice by treatment with tamoxifen for 4 days. Animals not treated with tamoxifen (including parental

mice) showed normal FKTN expression. As suggested by the reviewer, we have shown that tamoxifen treatment of 10-week-old α MHC-MCM-Fktn-cKO mice for 4 days induced chamber dilation and severe cardiac dysfunction with a ~50% death rate within 1 week (Fig. 6). Severe cardiac dysfunction is triggered by a decline in FKTN expression in tamoxifen-treated cKO mice (Supplementary Fig. 3c). In the revised manuscript, Fig. 6–8 show the results of our analyses of myocyte structure and function in tamoxifen-treated cKO mice. Supplementary Fig. 4 (middle panels) shows that the expression and localization of glycosylated α -DG and DGC proteins including α -DG at sarcolemma is normal in cKO hearts after 4 days of tamoxifen treatment and declines after 8 days of treatment. These observations suggest that the cardiac dysfunction observed after 4 days of tamoxifen treatment (Fig. 6b) is independent of α -DG glycosylation and DGC proteins in sarcolemma (p. 12). Therefore, we propose that a lack of FKTN protein *per se* affects myocyte structure and function in cKO hearts after 4 days of tamoxifen treatment and is the trigger for pathological remodeling of cKO hearts that is observed at 8 days post-treatment.

Further comment:

The authors responded to the question 2 by saying that “Fktn was eliminated in 10-week-old MHC-MCM-Fktn-cKO mice by treatment with tamoxifen for 4 days”. Where is the evidence that tamoxifen treatment can be so swift and effective to eliminate the FKTN protein within 4 days or even 7 days? What is the half life of FKTN mRNA and protein? the only data presented in the SI Fig 3c is the reduction of FKTN, probably at 50% of normal level by WB. Further, there is no description about how many days the samples were collected after Tamoxifen treatment (although described with 4 day Tam treatment) in the Figure legend. It is hard to image that all the described changes in MT and the death of mice can be attribute to 50% reduction of FKTN. Therefore, it is critical to demonstrate the changes in levels of FKTN within a week after Tamoxifen treatment if the authors want to claim that the observed phenomenon is the result of “elimination of the FKTN”. Without such evidence, all the conclusions can hardly be sustained.

Response: We checked FKTN expression after tamoxifen treatment and revised Supplementary Figure 3d with quantification of FKTN expression. In this study, we treated α MHC-MCM-Fktn-cKO mice with tamoxifen or vehicle for 4 consecutive days, and subsequently carried out analysis at different time points. We have added the scheme for experimental design in Supplementary Figure 3b, and revised the legends of supplementary Figures 3 and 4. FKTN protein expression showed about 80-90 % reduction in α MHC-MCM-Fktn-cKO mice after 4 days of tamoxifen treatment (4d) or 6 days after the onset of tamoxifen treatment (6d), compare with vehicle control (Supplementary Figure 3d). This data also suggest that the half-life of FKTN protein is less than 4 days.

Cardiac-specific and tamoxifen-inducible elimination of FKTN expression in α MHC-MCM-*Fktn*-cKO mice affected neither the embryonic development nor the growth after birth in the absence of tamoxifen treatment. Therefore, the acute elimination of FKTN function in α MHC-MCM-*Fktn*-cKO mice might circumvent any hypothetical compensatory process. We observed an obvious FKTN-deficiency phenotype in tamoxifen-treated α MHC-MCM-*Fktn*-cKO mice even young adult stage (Figure 6-8, and 10). In addition, the heart-failure phenotype in tamoxifen-treated α MHC-MCM-*Fktn*-cKO mice was ameliorated with colchicine treatment (Figure 9), suggesting that the maintenance of Golgi-microtubule networks was important for the preservation of cardiac function. Taken together, our study revealed that FKTN is crucial for maintaining myocyte physiology to prevent heart failure, and may lead to potential strategies for therapeutic intervention.

We added experimental details for FKTN detection in Materials and Methods.

To detect the expression of FKTN protein, cardiac muscles were solubilized with 1% Triton X-100 in TBS. The solubilized samples were protein assayed and then 4 mg of proteins were subjected to FKTN antibody-conjugated beads to enrich FKTN proteins as previously reported⁶². The bound materials were eluted with 0.1M Glycine-HCl (pH 2.5) and then analyzed by Western Blotting using FKTN antibody (RY213).

Reference

62. Sudo, A. et al., Temporal requirement of dystroglycan glycosylation during brain development and rescue of severe cortical dysplasia via gene delivery in the fetal stage. *Hum. Mol. Genet.* **1**, 1174-1185 (2018)

Reviewer #4 (Remarks to the Author):

The authors carried out detailed molecular and cellular analyses using two cardiac-specific, conditional *Fktn* knockout mouse models. They reported that microtubule densification may underlie the pathogenesis of cardiac dysfunction in the absence of *Fktn* and that treatment with microtubule depolymerization drug can improve the contractility in *Fktn*-deficient cardiomyocytes. I did not review the first version of the manuscript and was asked to comment on the response to Reviewer 3's comments. I think the authors have sufficiently addressed the questions raised by Reviewer 3.

Response: We thank for reviewer for his/her through consideration of our response to the comments of reviewer #3, and for his/her positive comments.

Reviewers' Comments:

Reviewer #2:

Remarks to the Author:

I remain deeply unconvinced that all the acute MT changes and cardiac failure described by the authors are the result of acute reduction in Fukutin expression (really limited from the initial WB, although the authors claimed to be >80% reduction). There is no sound evidence for such hypothesis.

Reviewer #5:

Remarks to the Author:

- The methods section states the mouse line crosses for this project.
- However, figure legends and results do not clearly state what genotype animals were examined. As such, the information is ambiguous. At times, the labels indicate that no cre control was used.
- Figure 1 – 4 labels indicate use of wrong control.
- Figure 5 labels are inadequate to determine whether appropriate controls were used.
- Figure 6, 7 labels suggest that the wrong controls were used, since the correct control should contain cre.
- Moreover, - colchicine should be indicated where appropriate, or, better yet, the vehicle that was used to dissolve colchicine.
- Figure 8 has a good number of wrong controls.
- Figure 10A clearly indicates that the wrong control was used, since the correct control should contain cre.
- Use of "KO" for conditional gene inactivation should be avoided, b/c this term is usually reserved for germline gene inactivation. Moreover, this term is misleading for conditional gene inactivation b/c cre lines usually do not have 100% gene inactivation efficiency. That is, tissues are not "KO", they may be "80% KO".

Responses to reviewers' comments

Reviewer #2 (Remarks to the Author):

I remain deeply unconvinced that all the acute MT changes and cardiac failure described by the authors are the result of acute reduction in Fukutin expression (really limited from the initial WB, although the authors claimed to be >80% reduction). There is no sound evidence for such hypothesis.

Response:

We added a new figure (Supplementary Fig. 5 in the revised manuscript) showing that tamoxifen treatment has no impact on the overall cardiac structure and function in *Fktn*^{flox/+};αMHC-MerCreMer (MCM)^{+/-} (hetero) mice. In addition, we did not observe any evidence for fibrosis and chamber dilation in the tamoxifen-treated *Fktn*^{flox/+};αMHC-MCM^{+/-} (hetero) hearts. Together, our results demonstrate that the heart-failure phenotype observed in αMHC-MCM-*Fktn*-cKO (*Fktn*^{flox/flox};αMHC-MCM^{+/-}) mice is not due to the adverse effect of tamoxifen treatment. In our study, we found an 80–90% reduction in FKTN protein expression in αMHC-MCM-*Fktn*-cKO (*Fktn*^{flox/flox};αMHC-MCM^{+/-}) mice after 4 days of tamoxifen treatment (4d) or 6 days after the onset of tamoxifen treatment (6d), compared with vehicle control (Supplementary Fig. 4d). Microarray analysis also showed that *Fktn* expression was reduced by approximately 70% in *Fktn*^{flox/flox};αMHC-MCM^{+/-} hearts treated with tamoxifen for 4 days (Fig. 10b). These data strongly suggest that severe cardiac dysfunction in tamoxifen-treated *Fktn*^{flox/flox}; αMHC-MCM^{+/-} mice was caused by acute reduction of FKTN protein expression.

In this study, we administered tamoxifen by intraperitoneal (i.p.) injection for four consecutive days at a dosage of 8 mg/kg body weight in 10-week-old mice. This dose was sufficient for the reduction of FKTN protein expression and *Fktn* transcription (Supplementary Fig. 4d and Fig. 10b). Consistent with the reviewer's viewpoint, several studies have raised concerns that tamoxifen-stimulated MerCreMer (MCM) in adult hearts may adversely influence cardiac function. Bersell *et al.*¹ have reported that injecting 3 × 30 mg/kg body weight of tamoxifen resulted in 10% mortality in 6-week-old αMHC-MCM^{+/-} mice. These hearts showed a DNA damage response, resulting in cardiomyocyte apoptosis, cardiac fibrosis, and cardiac dysfunction.¹ In our microarray analysis, we were unable to detect any alterations in the signaling pathways involved in DNA damage response in *Fktn*^{flox/flox}; αMHC-MCM^{+/-} hearts treated with tamoxifen for 4 days (Fig. 10). Hall *et al.*² have reported that administration of tamoxifen by i.p injection. for five consecutive days at 40 mg/kg leads to systolic dysfunction in αMHC-MCM^{+/-} mice. Therefore, Koitabashi *et al.*³ recommended that the tamoxifen dosage be reduced to less than 20 mg/kg per day for use in αMHC-MCM^{+/-} mice (an approximate total dose of 80 mg/kg). Thus, our study, which used a tamoxifen dose of 8 mg/kg

per day (an approximate total dose of 32 mg/kg), was designed to circumvent the adverse effects of tamoxifen and excessive MCM expression on cardiac function. In addition, cre-lines used in conditional knockouts usually are not 100% efficient for gene inactivation. Bersell *et al.*¹ have also reported that Cre-mediated homologous recombination at *loxP* in α MHC-MCM^{+/-} mice sites was dose-dependent and had a ceiling effect at ~80% of cardiomyocytes showing recombination. Thus, our observation of about 80–90% reduction in FKTN levels in *Fktn*^{fllox/fllox}; α MHC-MCM^{+/-} hearts after 4 days of tamoxifen treatment is consistent with the previous studies. Therefore, we believe that all the acute MT changes and cardiac failure observed in our study are the result of acute reduction in Fukutin expression. This discussion is also described in pages 28-29 of the revised manuscript.

References

1. Bersell K. et al., Moderate and high amounts of tamoxifen in α MHC-MerCreMer mice induce a DNA damage response, leading to heart failure and death. *Dis. Model. Mech.* **6**, 1459-1469 (2013).
2. Hall M.E. et al., Systolic dysfunction in cardiac-specific ligand-inducible MerCreMer transgenic mice. *Am. J. Physiol. Heart Circ. Physiol.* **301**, H253-H260 (2011)
3. Koitabashi N. et al., Avoidance of transient cardiomyopathy in cardiomyocyte-targeted tamoxifen-induced MerCreMer gene deletion models. *Circulation Research*, **105**, 12-15 (2009)

Reviewer #5 (Remarks to the Author):

- The methods section states the mouse line crosses for this project.
- However, figure legends and results do not clearly state what genotype animals were examined. As such, the information is ambiguous. At times, the labels indicate that no cre control was used.
- Figure 1 – 4 labels indicate use of wrong control.
- Figure 5 labels are inadequate to determine whether appropriate controls were used.
- Figure 6, 7 labels suggest that the wrong controls were used, since the correct control should contain cre.
- Figure 8 has a good number of wrong controls.
- Figure 10A clearly indicates that the wrong control was used, since the correct control should contain cre.

Response:

We thank the reviewer for his/her thorough consideration. As suggested by the reviewer, we have added new Supplementary figures (Supplementary Fig. 2 and Supplementary Fig. 5), which show

the use of cre-control, and the labels of the figures and the figure legends have been modified to clearly indicate the genotypes of the animals that were examined. In addition, we have provided a detailed description of the production of cKO mice and the background genotypes (pages 30–31 in the revised manuscript). Supplementary Fig. 2 illustrates that $Fktn^{lox/+};MCK-Cre^{+/-}$ (hetero) mice showed no abnormal cardiac structure and function at 48 weeks after birth. We did not observe any difference in cardiac structure and function between $Fktn^{lox/lox}$ (floxed control) and $Fktn^{lox/+};MCK-Cre^{+/-}$ (hetero) mice. These results suggest that the display of a heart-failure phenotype in $Fktn^{lox/lox};MCK-Cre^{+/-}$ (MCK-*Fktn*-cKO) mice at later stages (48 weeks) is dependent on FKTN protein expression.

MCK-promoter-driven transgenes are most highly expressed in skeletal muscle, and at lower levels in cardiac muscle.⁴ Therefore, there is little possibility of expressing excessive levels of cre-recombinase in the cardiac muscle of MCK-Cre mice. Cardiac abnormality has never been reported in MCK-Cre^{+/-} mice. In our study, to delete the floxed alleles in the cardiac muscle, $Fktn^{lox/lox}$ mice were mated with transgenic mice expressing cre-recombinases under the control of MCK promoter (B6.FVB(129S4)-Tg(Ckmm-cre)5Khn/J; The Jackson Laboratory), and the offspring were back-crossed with the $Fktn^{lox/lox}$ mice (C57BL/6 background) over 25 generations (see page 30 in revised manuscript). Therefore, $Fktn^{lox/lox}$ littermates were used as age-matched controls in this study. In several studies including ours, floxed mice were used as controls with the same genetic background^{5,6}. Although we understand that both floxed-homozygous ($Fktn^{lox/lox}$) mice and floxed-heterozygous mice with MCK-Cre^{+/-} ($Fktn^{lox/+};MCK-Cre^{+/-}$) should ideally be used in all experiments, we selected the floxed mice as our control because of the recessive genetic traits of muscular dystrophy and cardiomyopathy in FKTN-deficient mice have been reported in previous studies^{5,6}. This strategy also enabled us to reduce the number of experimental animals used in this study.

We have added a new figure (Supplementary Fig. 5 in the revised manuscript) showing that tamoxifen treatment has no impact on cardiac structure and function in $Fktn^{lox/+};\alpha MHC-MCM^{+/-}$ (hetero) mice, thus demonstrating that the heart-failure phenotype observed in tamoxifen-treated $\alpha MHC-MCM-Fktn$ -cKO ($Fktn^{lox/lox};\alpha MHC-MCM^{+/-}$) mice is not due to the adverse effects of tamoxifen treatment. Based on these results and previous literature that reports the recessive genetic traits of muscular dystrophy and cardiomyopathy in FKTN-deficient mice^{5,6}, we strongly believe that our use of floxed-homozygous mice as controls in this study was appropriate. This discussion is also added on pages 28–29 of the revised manuscript.

As shown in Fig. 6a and Supplementary Fig. 5, no abnormalities in cardiac morphology and function

were observed in *Fktn*^{fllox/fllox} (floxed control) and *Fktn*^{fllox/+};αMHC-MCM^{+/-} (hetero) mice with or without tamoxifen treatment. In addition, vehicle-treated *Fktn*^{fllox/fllox};αMHC-MCM^{+/-} hearts also showed normal cardiac structure and function. Therefore, the effects of FKTN loss on cellular physiology were examined by comparing myocytes isolated from *Fktn*^{fllox/fllox};αMHC-MCM^{+/-} hearts with or without tamoxifen-treatment. Thus, we examined the effects of loss of FKTN in isolated myocytes with the same genetic background.

In our study, αMHC-MCM^{+/-} mice (Stock No. 005650, FVB/N background; The Jackson Laboratory) were back-crossed with C57BL/6 mice over 20 generations (see page 30 in revised manuscript). To delete the floxed alleles in the cardiac muscle, *Fktn*^{fllox/fllox} mice were mated with αMHC-MCM^{+/-} mice from obtained colony, and the offspring were backcrossed with the *Fktn*^{fllox/fllox} mice (C57BL/6 background) over 20 generations (see method in revised manuscript). Therefore, *Fktn*^{fllox/fllox} littermates were used as age-matched controls in this study. Although we understand that both floxed-homo (*Fktn*^{fllox/fllox}) mice and *Fktn*^{fllox/+};αMHC-MCM^{+/-} (hetero) mice with or without tamoxifen should ideally be used in all experiments, we selected the floxed mice as control because of the recessive genetic traits of muscular dystrophy and cardiomyopathy in FKTN-deficient mice reported in previous studies^{5,6}. This strategy enabled us to reduce the number of experimental animals used in this study, and may have enhanced the sensitivity of detecting changes in gene expression.

- Moreover, colchicine should be indicated where appropriate, or, better yet, the vehicle that was used to dissolve colchicine.

We added the schedule of tamoxifen administration, colchicine treatment, and sampling in Fig. 9a. In addition, we added the explanation for the vehicle used, in the Methods section (p. 31).

- Use of “KO” for conditional gene inactivation should be avoided, b/c this term is usually reserved for germline gene inactivation. Moreover, this term is misleading for conditional gene inactivation b/c cre lines usually do not have 100% gene inactivation efficiency. That is, tissues are not "KO", they may be "80% KO".

We understand what this reviewer suggested. Cre-lines used in conditional knockout usually do not have 100% gene inactivation efficiency. Bersell *et al.*¹ have also reported that Cre-mediated homologous recombination at *loxP* sites was dose-dependent and had a ceiling effect at ~80% of cardiomyocytes showing recombination. These observations reinforce that about 80-90 % reduction in *Fktn*^{fllox/fllox}; αMHC-MCM^{+/-} hearts after 4 days of tamoxifen-treatment is reasonable. Thus, we

believe that all the acute MT changes and cardiac failure observed in our study are the result of acute reduction in Fukutin expression. We did not use the term “KO” for representing conditional gene inactivation in this manuscript. The term KO used in the text and figures represents HEK cells whose FKTN gene was disrupted by genome editing.

References

4. Nguyen Q.V. et al., Transgenic and tissue culture analysis of the muscle creatine kinase enhancer Trex control element in skeletal and cardiac muscle indicate differences in gene expression between muscle types. *Transgenic Research*, **12**, 337-349 (2003)
5. Kim Y.C. et al., Rag GTPases are cardioprotective by regulating lysosomal function. *Nature Communications*, **5**, 4241 (2014)
6. Mora A. et al., Deficiency of PKD1 in cardiac muscle results in heart failure and increased sensitivity to hypoxia. *EMBO J.* **22**, 4666-4676 (2003)

Reviewers' Comments:

Reviewer #2:

Remarks to the Author:

No

Reviewer #5:

Remarks to the Author:

This reviewer disagrees with this argument provided in the rebuttal letter: "because of the recessive genetic traits of muscular dystrophy and cardiomyopathy in FKTN-deficient mice reported in previous studies^{5,6}."

- This argument works to the disadvantage of the authors, i.e., the alpha-MHC-MerCreMer^{+/-}; FKTN^{flox/wt} mice should be used as controls, because of the heterozygous phenotype.

The tamoxifen dose used was 8 mg/kg x 4 days. This dose is approximately 1/3 of the dose used by most cardiac labs, and yet there is approximately 80% protein reduction in 4 days. This reviewer shares the concern of another reviewer about the FKTN protein reduction within 4 days, which is super-fast. The only way this could be explained is for FKTN mRNA and/or FKTN protein to have a very fast turnover rate.

Suppl Fig 4D still has the previously criticized non-specific designation of CKO. This has to be fixed.

Suppl fig 6 still shows the wrong control.

Reviewer #6:

Remarks to the Author:

Several types of MDs are associated with severe cardiac defects that can be the main cause of death. Therefore, investigating the pathogenesis of cardiac defects in MD is a critical area of research. Using conditional mouse models, Ujihara et al. demonstrate that the loss of Fktn leads to a dramatic cardiomyopathy. The mouse models used to study the role of Fktn in the heart are appropriate and the in vivo experiments include the proper controls. The disruption of the MT network and Golgi fragmentation are two main pathological features of the cardiomyopathy described in Fktn-cKO mice. Work by other groups describes very similar heart defects in different forms of MD and it is unclear which pathomechanisms revealed in Fktn-cKO mice are unique to this form of MD. The authors treat Fktn-cKO animals with colchicine and that improved cardiac function. However as pointed out in the discussion, treating mdx mice with colchicine was also beneficial for the heart by correcting t-tubule architecture in these mice. Overall, the authors performed a tremendous amount of work, however as a whole, the data lack cohesiveness and novelty.

Specific Comments:

1. The abstract should be revised to better reflect the data presented in the manuscript. For example, the second line of the abstract reads as follows 'we show the detailed molecular pathogenesis of MD-associated cardiomyopathy using mice lacking the fukutin gene'. This suggests that the molecular pathogenesis uncovered in the study can be generalized to all MDs which is likely not the case.

2. Also in the abstract, the authors state that 'Fktn-deficient mice were vulnerable to pathological hypertrophic stress via impaired activation of Akt and the MEF2-histone deacetylase axis'. This is

an overstatement; the data are purely correlative and no causation was established.

3. Is tubulin an appropriate loading control for Supplementary Figure 1 given the data in Figure 7?

4. Keramaris et al. (2006, *Muscle&Nerve*) show robust I1H6 staining beginning at 4 weeks of age in the heart. How can the authors explain the discrepancy between what is reported in the literature and their findings in Figure 1 and Supplementary Figure 6 with regards to the immunostaining for the glycosylated form of α -DG?

5. In Figure 1, the authors report a substantial increase in cardiomyocyte size without a corresponding increase in h.w./b.w. ratio. These data are suggestive of cardiomyocyte dropout in MCK-fktn-cKO mice. Perhaps it would be beneficial to demonstrate that the integrity of the sarcolemma is compromised leading to cardiomyocyte death. This could possibly be achieved by immunostaining for IgM or even staining the slides with Evan's blue dye (in addition to assessing cell death in the hearts).

6. It would appear that the authors have mistakenly used the same immunofluorescence image in 2 separate figures. In Figure 2A, the Fktnflox/flox LTCC immunofluorescence image is also found in Supplementary Figure 3 but this time under a different genotype (Fktnflox/flox;MCK-Cre+/-).

7. On page 8 - line 138, the authors state that isoproterenol treatment induced the phosphorylation of PKD. This reviewer finds it difficult to see the increase in PKD phosphorylation in Figure 3F. Furthermore, the authors believe that the lower PKD band observed in the MCK-fktn-cKO samples is a PKD proteolysis product. A more plausible explanation for the lower band is that the antibody is not specific to PKD and also picks up PKD2. In fact, the Cell Signaling data sheet that accompanies the antibody indicates that the antibody can react with PKD2 which is 10 kDa smaller than PKD. The appearance of PKD2 is likely attributable to the increased level of other cell types like fibroblasts and or immune cells in hearts of MCK-fktn-cKO mice following isoproterenol treatment.

8. Are the baseline changes reported in Figure 3F also present in MCK-fktn-cKO mice at 48 weeks?

9. The scale bar is wrong in either Figure 3A or Supplementary Figure 2A.

10. The authors should show that Fktn is deleted from the cardiomyocytes used in Figure 4. Assuming that the gene is deleted, these cardiomyocytes have a significant phenotype at baseline even though no defects are reported in younger MCK-fktn-cKO mice. Can the authors comment on how this could be?

11. The authors should indicate the time point used for the given experiments in each figure legend.

12. The authors devote an entire figure to the Golgi defects observed in α MHC-MCK-fktn-cKO mice. However, Golgi fragmentation in heart failure has previously been reported and the link between MTs and the localization of the Golgi apparatus is very well established. Frankly, Figure 10 does not really add much to the paper. In fact, treating the mice with colchicine improved heart function in α MHC-MCK-fktn-cKO mice despite the Golgi fragmentation.

13. The data presented in Figure 5d are not convincing enough to suggest Golgi fragmentation under basal conditions.

14. The data from Supplementary Figure 10 should be discussed in the results section. The findings in Supplementary Figure 10 are not overly convincing and likely require higher resolution

images to support the stated conclusion.

Responses to reviewers' comments:

Reviewer#5 (Remarks to the Author):

This reviewer disagrees with this argument provided in the rebuttal letter: “because of the recessive genetic traits of muscular dystrophy and cardiomyopathy in FKTN-deficient mice reported in previous studies^{5,6}.”

- This argument works to the disadvantage of the authors, i.e., the α -MHC-MerCreMer^{+/-}; FKTN^{flox/wt} mice should be used as controls, because of the heterozygous phenotype.

We thank the reviewer for this important comment. We used heterozygous control mice ($Fktn^{flox/+}$;MCK-Cre^{+/-} and tamoxifen-treated $Fktn^{flox/+}$; α MHC-MCM^{+/-}) and found that these mice do not exhibit any abnormalities in cardiac structure, function, or survival probability (Supplementary Fig. 3 and Supplementary Fig. 7 in this revision). In addition, we added new data to demonstrate that there are no differences in expression or localization of α DG (IHH6), β DG, α SG, β SG, and γ SG in the hearts of heterozygous ($Fktn^{flox/+}$; α MHC-MCM^{+/-}) mice with or without tamoxifen treatment (Supplementary Fig. 8c).

The tamoxifen dose used was 8 mg/kg x 4 days. This dose is approximately 1/3 of the dose used by most cardiac labs, and yet there is approximately 80% protein reduction in 4 days. This reviewer shares the concern of another reviewer about the FKTN protein reduction within 4 days, which is super-fast. The only way this could be explained is for FKTN mRNA and/or FKTN protein to have a very fast turnover rate.

We agree with this comment by the reviewer. We added new data to show that Cre recombination produced *Fktn* mRNA in $Fktn^{flox/flox}$; α MHC-MCM^{+/-} hearts treated with tamoxifen for 4 d (Supplementary Fig. 6d), suggesting that *Fktn* recombination had already occurred in the genome of these hearts. After 4 d of tamoxifen treatments, the original form of *Fktn* mRNA was reduced by about 80% in the hearts (Supplementary Fig. 7c). In addition, we found an 80–90% reduction in FKTN protein expression in α MHC-MCM-*Fktn*-cKO ($Fktn^{flox/flox}$; α MHC-MCM^{+/-}) mice after 4 d of tamoxifen treatment or 6 d after the onset of tamoxifen treatment compared with levels after vehicle control (Supplementary Fig. 6e). These data suggest that the half-life of FKTN protein is less than 4 d. Thus, FKTN mRNA and/or FKTN protein likely have a fast turnover rate. This description was included on page 30–31 of the revised manuscript.

Suppl Fig 4D still has the previously criticized non-specific designation of CKO. This has to be fixed.

We thank the reviewer for pointing this error out. We modified the designation of “cKO” in the label of Supplementary Fig. 6e in the revised manuscript (previously Supplementary Fig. 4d in the previous manuscript).

Suppl fig 6 still shows the wrong control.

We thank the reviewer for pointing this out. We made the necessary change in Supplementary Fig. 8. We added new data (Supplementary Fig. 6 in the previous manuscript) to demonstrate that tamoxifen treatment has no effect on changes in expression and localization of α DG (IH6), β DG, α SG, β SG, and γ SG in the hearts of floxed-*Fktn* (*Fktn*^{flox/flox}) and heterozygous (*Fktn*^{flox/+}; α MHC-MCM^{+/-}) mice with or without tamoxifen treatment. In addition, we did not observe any evidence for cardiomyocyte degeneration in the tamoxifen-treated *Fktn*^{flox/flox} (floxed-*Fktn*) or *Fktn*^{flox/+}; α MHC-MCM^{+/-} (hetero) hearts. Together, our results demonstrate that the heart-failure phenotype observed in α MHC-MCM-*Fktn*-cKO (*Fktn*^{flox/flox}; α MHC-MCM^{+/-}) mice is not due to the adverse effect of tamoxifen treatment.

Reviewer #6 (Remarks to the Author):

Several types of MDs are associated with severe cardiac defects that can be the main cause of death. Therefore, investigating the pathogenesis of cardiac defects in MD is a critical area of research. Using conditional mouse models, Ujihara et al. demonstrate that the loss of *Fktn* leads to a dramatic cardiomyopathy. The mouse models used to study the role of *Fktn* in the heart are appropriate and the in vivo experiments include the proper controls. The disruption of the MT network and Golgi fragmentation are two main pathological features of the cardiomyopathy described in *Fktn*-cKO mice. Work by other groups describes very similar heart defects in different forms of MD and it is unclear which pathomechanisms revealed in *Fktn*-cKO mice are unique to this form of MD. The authors treat *Fktn*-cKO animals with colchicine and that improved cardiac function. However as pointed out in the discussion, treating mdx mice with colchicine was also beneficial for the heart by correcting t-tubule architecture in these mice. Overall, the authors performed a tremendous amount of work, however as a whole, the data lack cohesiveness and novelty.

Specific Comments:

1. The abstract should be revised to better reflect the data presented in the manuscript. For example, the second line of the abstract reads as follows 'we show the detailed molecular pathogenesis of MD-associated cardiomyopathy using mice lacking the fukutin gene'. This suggests that the molecular pathogenesis uncovered in the study can be generalized to all MDs which is likely not the case.

We agree with the reviewer and revised the abstract accordingly.

2. Also in the abstract, the authors state that 'Fktn-deficient mice were vulnerable to pathological hypertrophic stress via impaired activation of Akt and the MEF2-histone deacetylase axis'. This is an overstatement; the data are purely correlative and no causation was established.

We thank the reviewer for this comment, and we revised the abstract in accord with this comment.

3. Is tubulin an appropriate loading control for Supplementary Figure 1 given the data in Figure 7?

In response to this question, we added a new panel showing the membrane stained with ponceau-S to serve as a loading control in Supplementary Fig. 1 in the revised manuscript.

4. Keramaris et al. (2006, Muscle&Nerve) show robust I1H6 staining beginning at 4 weeks of age in the heart. How can the authors explain the discrepancy between what is reported in the literature and their findings in Figure 1 and Supplementary Figure 6 with regards to the immunostaining for the glycosylated form of α -DG?

We thank the reviewer for this comment. As the reviewer pointed out, Keramaris et al. showed robust I1H6 reaction beginning at 4 weeks of age in the heart (Muscle & Nerve, 2017). According to the details of the immunohistochemistry as described in their paper, the cardiac cross-section was blocked with 8% bovine serum albumin (BSA) in ddH₂O for 1 h. However, the anti-I1H6 antibody is a mouse monoclonal antibody. Therefore, in our study, we used Mouse On Mouse (M.O.M.) Blocking Reagent (Vector Laboratories) to block endogenous mouse immunoglobulins in the mouse hearts in our study (Fig. 1). Anti-I1H6 antibody was tested at a dilution of 1:500 (Keramaris et al. also used the antibody at a 1:500 dilution). In hearts from 4-week-old mice, we also detected robust I1H6 staining using 8% BSA blocking solution (bottom panels); however, we did not detect this robust I1H6 signal in hearts blocked with M.O.M. Blocking Reagent. Thus, we conclude that the discrepancy between the result reported in the literature by Keramaris et al. and that in our study is caused by the difference in blocking solution. This blocking protocol is described on page 40 of the

revised manuscript.

5. In Figure 1, the authors report a substantial increase in cardiomyocyte size without a corresponding increase in h.w./b.w. ratio. These data are suggestive of cardiomyocyte dropout in MCK-*fkn*-cKO mice. Perhaps it would be beneficial to demonstrate that the integrity of the sarcolemma is compromised leading to cardiomyocyte death. This could possibly be achieved by immunostaining for IgM or even staining the slides with Evan's blue dye (in addition to assessing cell death in the hearts).

As the reviewer suggested, the cross-sectional area was increased in MCK-*Fkn*-cKO hearts 48 weeks after birth (Fig. 1e), suggesting cardiomyocyte dropout in the hearts. In contrast, MCK-*Fkn*-cKO hearts at 48 weeks did not show this significant change in the HW/BW ratio. This difference is likely due to initiation of fibrosis following cardiomyocyte dropout (Fig. 1f).

As for the hypothesis that the integrity of the sarcolemma is compromised, leading to cardiomyocyte death, we added new data to illustrate the cell damage in MCK-*Fkn*-cKO hearts at 48 weeks after birth (Supplementary Fig. 2). We described these results on page 6, line 1–2. MCK-*Fkn*-cKO (*Fkn*^{fllox/fllox};MCK-*Cre*^{+/+}) hearts showed clustered patches of IgG-positive myocytes in-contact with neighboring cells at 48 weeks after birth; however, the damaged areas of the hearts were limited to

only a single or a few places in these 48-week-old MCK-*Fktn*-cKO mice. In contrast, cardiac dysfunction was observed from 24 weeks of age in these mice (Fig. 1g, h). Thus, we believe that the cardiac dysfunction in the hearts of MCK-*Fktn*-cKO mice 24–48 weeks after birth is mainly due to the decline in the myocyte contractile function, as shown in Fig. 2.

6. It would appear that the authors have mistakenly used the same immunofluorescence image in 2 separate figures. In Figure 2A, the *Fktn*^{flox/flox} LTCC immunofluorescence image is also found in Supplementary Figure 3 but this time under a different genotype (*Fktn*^{flox/flox};MCK-Cre^{+/-}).

We appreciated the reviewer's comment. We corrected this mistake in Supplementary Fig. 4a in the revised manuscript (previously Supplementary Fig. 3a).

7. On page 8 - line 138, the authors state that isoproterenol treatment induced the phosphorylation of PKD. This reviewer finds it difficult to see the increase in PKD phosphorylation in Figure 3F. Furthermore, the authors believe that the lower PKD band observed in the MCK-*fktn*-cKO samples is a PKD proteolysis product. A more plausible explanation for the lower band is that the antibody is not specific to PKD and also picks up PKD2. In fact, the Cell Signaling data sheet that accompanies the antibody indicates that the antibody can react with PKD2 which is 10 kDa smaller than PKD. The appearance of PKD2 is likely attributable to the increased level of other cell types like fibroblasts and or immune cells in hearts of MCK-*fktn*-cKO mice following isoproterenol treatment.

We thank the reviewer for this comment. We replaced the S916-PKD panel of Fig. 3F with an image with a longer exposure. In addition, we modified the text in the revised manuscript (page 8, lines 17–18). Anti-phospho PKD (S916) and -phospho PKD (S744) antibodies recognized two bands (about 120 kDa and 90 kDa) in isoproterenol-treated MCK-*Fktn*-cKO hearts (Supplementary Fig. 14a). We calculated the intensity of the upper bands (about 120 kDa) to generate the graph. The molecular weight of PKD2 is about 105 kDa, and the lower band was 90 kDa. However, as we do not have direct data to demonstrate that the 90-kDa bands arose from proteolysis, we deleted the description on proteolysis from the text.

8. Are the baseline changes reported in Figure 3F also present in MCK-*fktn*-cKO mice at 48 weeks?

We added new data (Supplementary Fig. 11 in the revised manuscript) to show the phosphorylation

levels of PKD and Akt in MCK-*Fktn*-cKO mice at 48 weeks. Although the phosphorylated form of PKD was similar to that in the control hearts, total PKD protein levels were slightly increased in MCK-*Fktn*-cKO mice at 48 weeks. Therefore, the phosphorylated level was reduced in MCK-*Fktn*-cKO mice at 48 weeks. On the other hand, the phosphorylation level of Akt was clearly downregulated in MCK-*Fktn*-cKO mice at 48 weeks. Thus, both the isoproterenol-treated MCK-*Fktn*-cKO hearts at 10 weeks (young adult) and MCK-*Fktn*-cKO hearts at 48 weeks (old age) presented cardiac dysfunction with disturbed PKD and Akt. These symptoms appear to reflect the pathogenesis and/or pathological consequences depending on the timing of *Fktn* deletion, disease progression course (period), and hemodynamic stress condition. In both cases, however, our data showed that FKTN protein affects the maintenance of the Golgi-MT network in cardiomyocytes. With respect to this interpretation and possible explanation, we modified the statements in the Discussion (from page 26, line 1–13).

9. The scale bar is wrong in either Figure 3A or Supplementary Figure 2A.

We appreciated the reviewer's comment. We corrected the mistake in Supplementary Fig. 3A (previously Supplementary Fig. 2A).

10. The authors should show that *Fktn* is deleted from the cardiomyocytes used in Figure 4. Assuming that the gene is deleted, these cardiomyocytes have a significant phenotype at baseline even though no defects are reported in younger MCK-*fktn*-cKO mice. Can the authors comment on how this could be?

We thank the reviewer for this comment. We added new data showing that Cre-lox recombination had already occurred in neonatal MCK-*Fktn*-cKO cardiomyocytes cultured for 48 h in Supplementary Fig. 5 in the revised manuscript. We also showed that the neonatal cardiomyocytes isolated from MCK-*Fktn*-cKO mice expressed Cre-recombinase in the nucleus (Supplementary Fig. 5). These observations suggest that the reduction of FKTN protein affects the hypertrophic response in cultured myocytes isolated from neonatal hearts. We also confirmed reduction of FKTN protein in 1-d-old MCK-*Fktn*-cKO mice (Supplementary Fig. 5 and page 9, line 13–16).

As shown in Fig. 4, the PE-treated *Fktn*-deficient cardiomyocytes exhibited a severe phenotype. Cardiomyocyte culture with PE is a well-established and widely used experimental model for the assessment of maturation and hypertrophic response. Indeed, our data suggest that *Fktn* elimination in myocytes impairs hypertrophic response (Fig. 4 and Fig. 5a, b), consistent with our *in vivo* data that MCK-*Fktn*-cKO hearts show vulnerability to pathological hypertrophic stress (Fig. 3).

Younger MCK-*Fktn*-cKO mice have no abnormality in cardiac structure or function under physiological conditions; however, the cell areas of neonatal cardiomyocytes from MCK-*Fktn*-cKO mice cultured with or without PE are clearly smaller than those from control hearts (Fig. 4c), suggesting an abnormality. Unfortunately, the primary cardiomyocyte culture system is not applicable to assess physiological cardiac function. Thus, we cannot directly compare the *in vitro* data gleaned from neonatal cultured cardiomyocyte and *in vivo* data obtained from studies of younger MCK-*Fktn*-cKO under physiological conditions. We provided a discussion of possible reasons why younger MCK-*Fktn*-cKO do not show any pathological phenotype under physiological conditions, and these possibilities include the presence of a compensatory mechanism (page 23, line 10 to page 24, line 4).

11. The authors should indicate the time point used for the given experiments in each figure legend.

We modified the figure legends.

12. The authors devote an entire figure to the Golgi defects observed in α MHC-MCM-*fkt*n-cKO mice. However, Golgi fragmentation in heart failure has previously been reported and the link between MTs and the localization of the Golgi apparatus is very well established. Frankly, Figure 10 does not really add much to the paper. In fact, treating the mice with colchicine improved heart function in α MHC-MCM-*fkt*n-cKO mice despite the Golgi fragmentation.

We appreciated the reviewer's comment. While our original version of this manuscript did not include Fig. 10 (and the associated data), we added this figure in response to the recommendation of one of the original reviewers. Fig. 10 clearly demonstrates that the reduction of FKTN affects Golgi structure and function.

13. The data presented in Figure 5d are not convincing enough to suggest Golgi fragmentation under basal conditions.

We appreciated the reviewer's comment. As reviewer suggested, it is difficult to judge the Golgi fragmentation in island-like beating myocytes (Fig. 5d in previous manuscript). Therefore, we replaced the panel that showed the localization of GM130 in FKTN-deficient neonatal myocytes isolated from MCK-*Fktn*-cKO mice (Fig. 5d in revised manuscript).

14. The data from Supplementary Figure 10 should be discussed in the results section. The findings in Supplementary Figure 10 are not overly convincing and likely require higher resolution images to support the stated conclusion.

We thank the reviewer for this comment. We confirmed that previous image has the saturation artefacts. Therefore, we replaced another 3D overlapping image in Supplementary Fig. 10a. In addition, the data were discussed in the Results section (page 16, line 17 to page 17, line 11). We also mentioned these data in the Discussion (page 26).

Reviewers' Comments:

Reviewer #5:

Remarks to the Author:

The rebuttal still presents possible effects of tamoxifen alone and does not appreciate are toxicity.

Reviewer #6:

Remarks to the Author:

revisions were adequate. I have no additional concerns

Responses to reviewers' comments:

We would like to thank the reviewers for their careful consideration of our manuscript.

Reviewer #5 (Remarks to the Author):

The rebuttal still presents possible effects of tamoxifen alone and does not appreciate are toxicity.

We believe that our revision appropriately addresses concerns on tamoxifen toxicity. We are happy to hear that both editors and referee 6 were satisfied with our revisions.

Reviewer #6 (Remarks to the Author):

revisions were adequate. I have no additional concerns

Thank you for your careful consideration of our manuscript. We appreciated the reviewer's comment.